# Controllability boosts neural and cognitive signatures of changes-of-mind in uncertain environments

Marion Rouault[1,2,3]*, Aurélien Weiss[1,3,4], Junseok K Lee[1,3], Jan Drugowitsch[5], Valerian Chambon[2,3†], Valentin Wyart[1,3*†]

[1]Laboratoire de Neurosciences Cognitives et Computationnelles, Institut National de la Santé et de la Recherche Médicale (Inserm), Paris, France; [2]Institut Jean Nicod, Centre National de la Recherche Scientifique (CNRS), Paris, France; [3]Département d'Études Cognitives, École Normale Supérieure, Université Paris Sciences et Lettres (PSL University), Paris, France; [4]Université de Paris, Paris, France; [5]Department of Neurobiology, Harvard Medical School, Boston, United States

**Abstract** In uncertain environments, seeking information about alternative choice options is essential for adaptive learning and decision-making. However, information seeking is usually confounded with changes-of-mind about the reliability of the preferred option. Here, we exploited the fact that information seeking requires control over which option to sample to isolate its behavioral and neurophysiological signatures. We found that changes-of-mind occurring with control require more evidence against the current option, are associated with reduced confidence, but are nevertheless more likely to be confirmed on the next decision. Multimodal neurophysiological recordings showed that these changes-of-mind are preceded by stronger activation of the dorsal attention network in magnetoencephalography, and followed by increased pupil-linked arousal during the presentation of decision outcomes. Together, these findings indicate that information seeking increases the saliency of evidence perceived as the direct consequence of one's own actions.

*For correspondence:
marion.rouault@gmail.com (MR);
valentin.wyart@gmail.com (VW)

†These authors contributed equally to this work

## Editor's evaluation

This article will be of interest to psychologists and cognitive neuroscientists studying learning, decision-making, belief formation, and metacognition. The authors use an elegant task in which people make decisions with or without control over the information they sample, and link the cognitive processes at play to magnetoencephalography and pupillometry signatures. The key finding is that when participants have control over information sampling (i.e., they are seeking information), they need more contradictory evidence in order to switch their choices, and such switches are made with lower confidence, which is a clear conceptual advance in this field.

## Introduction

The ability to form and revise uncertain beliefs through information sampling is a hallmark of human cognition. This inference process has been extensively studied using two main classes of decision tasks (*Bartolo and Averbeck, 2021*; *Wyart and Koechlin, 2016*). In passive sampling tasks, participants are observers who sample information over which they have no control (*Murphy et al., 2016*; *van den Berg et al., 2016*; *Zylberberg et al., 2018*). This is the case in most perceptual decision tasks, in which the experimenter controls the sensory information provided to participants (for reviews, see

*Gold and Shadlen, 2007*; *Hanks and Summerfield, 2017*). Outside the laboratory, for example, one might see an ad for a new movie at a bus stop. By contrast, active sampling tasks let participants choose which source of information to sample from *Charpentier et al., 2018*; *Gureckis and Markant, 2012*; *Markant and Gureckis, 2014*. In these situations, the information provided to participants corresponds to the outcome of their own choices. Using the same example, one can alternatively browse the web for information about new movies. Critically, unlike passive sampling, active sampling provides participants with control over which source of information to sample from, even if the information itself (about the new movie) can be exactly the same in both cases.

In both passive and active sampling tasks, recent work has assigned a key role for confidence in the formation and revision of uncertain beliefs (*Meyniel et al., 2015*; *Nassar et al., 2010*; *Rouault et al., 2019*; *Sarafyazd and Jazayeri, 2019*). Low confidence in current beliefs has been shown to predict changes-of-mind (*Balsdon et al., 2020*; *Folke et al., 2016*) and allows for the flexible adaptation of behavioral strategies even when external feedback is unavailable (*Desender et al., 2018*; *Desender et al., 2019*; *Fleming et al., 2018*; *Rollwage et al., 2020*). But besides this pervasive role of confidence in belief updating, the control conferred by active sampling allows one to seek information about alternative strategies, a cognitive process that is by definition not possible in the absence of control over information sampling. Information seeking has been mostly studied using 'exploration-exploitation' dilemmas (*Costa et al., 2019*; *Daw et al., 2006*), where it is confounded with changes-of-mind about the reliability of the current behavioral strategy. In these paradigms, participants evolve in controllable environments and usually sample one among several options to maximize reward. Therefore, they have to either exploit a currently rewarding option or sacrifice rewards to explore alternative options and seek information about their possible rewards (*Rich and Gureckis, 2018*; *Wilson et al., 2014*). This trade-off means that in these paradigms exploration differs from exploitation not only in terms of information seeking, but also in terms of other co-occurring cognitive events, including overt response switches and covert changes-of-mind.

These different families of paradigms developed for studying information seeking vary on several dimensions, particularly the sources of uncertainty (*Fleming et al., 2018*), stimuli used (*Gesiarz et al., 2019*), desirability of the information to be sought (*Hertwig and Engel, 2021*), and degree of control over information sampled (*Desender et al., 2018*). These differences have made direct comparisons between paradigms extremely challenging. To date, no study has directly manipulated control over evidence sampling in otherwise aligned experimental conditions. At the neurophysiological level, exploration is known to be associated with larger pupil-linked arousal (*Jepma and Nieuwenhuis, 2011*) and increased activity in lateral prefrontal regions in electroencephalographic (EEG) and blood oxygen-level dependent (BOLD) activity (*Donoso et al., 2014*; *Tzovara et al., 2012*). However, due to confounds between information seeking and other cognitive events during exploration in these studies, it remains unclear whether information seeking is associated with specific neurophysiological signatures.

Here, to isolate the specific signatures of information seeking, we contrasted changes-of-mind occurring with and without control over information sampling. We employed an adaptive decision-making task that allows comparing the formation and revision of uncertain beliefs between controllable and uncontrollable conditions (*Figure 1*). We previously used these experimental conditions to compare how participants integrate evidence when it is a cue (uncontrollable condition) vs. when it is an outcome (controllable condition) (*Weiss et al., 2021*). Here, we focus on the comparison of repeat and switch decisions between these conditions so as to isolate the behavioral signatures and neural basis of information seeking, while replicating most of the previously observed effects in Weiss et al. on behavioral choices and their computational modeling (*Weiss et al., 2021*). We found that participants need more evidence against their current beliefs to change their behavior in the controllable condition, and that they do so with decreased confidence. Nevertheless, participants are more likely to probe again their new behavioral strategy in the next decision, even in the absence of objective evidence supporting this new strategy – a form of active 'hypothesis testing' (*Markant and Gureckis, 2014*). Using computational modeling (*Glaze et al., 2015*; *Weiss et al., 2021*), we show that controllability increases the stability of participants' beliefs, a mechanism that explains the observed behavioral correlates of information seeking. At the physiological level, changes-of-mind occurring in the controllable condition are preceded by stronger suppression of neuromagnetic alpha-band activity in the dorsal attention network, and followed by increased pupil-linked arousal during the presentation

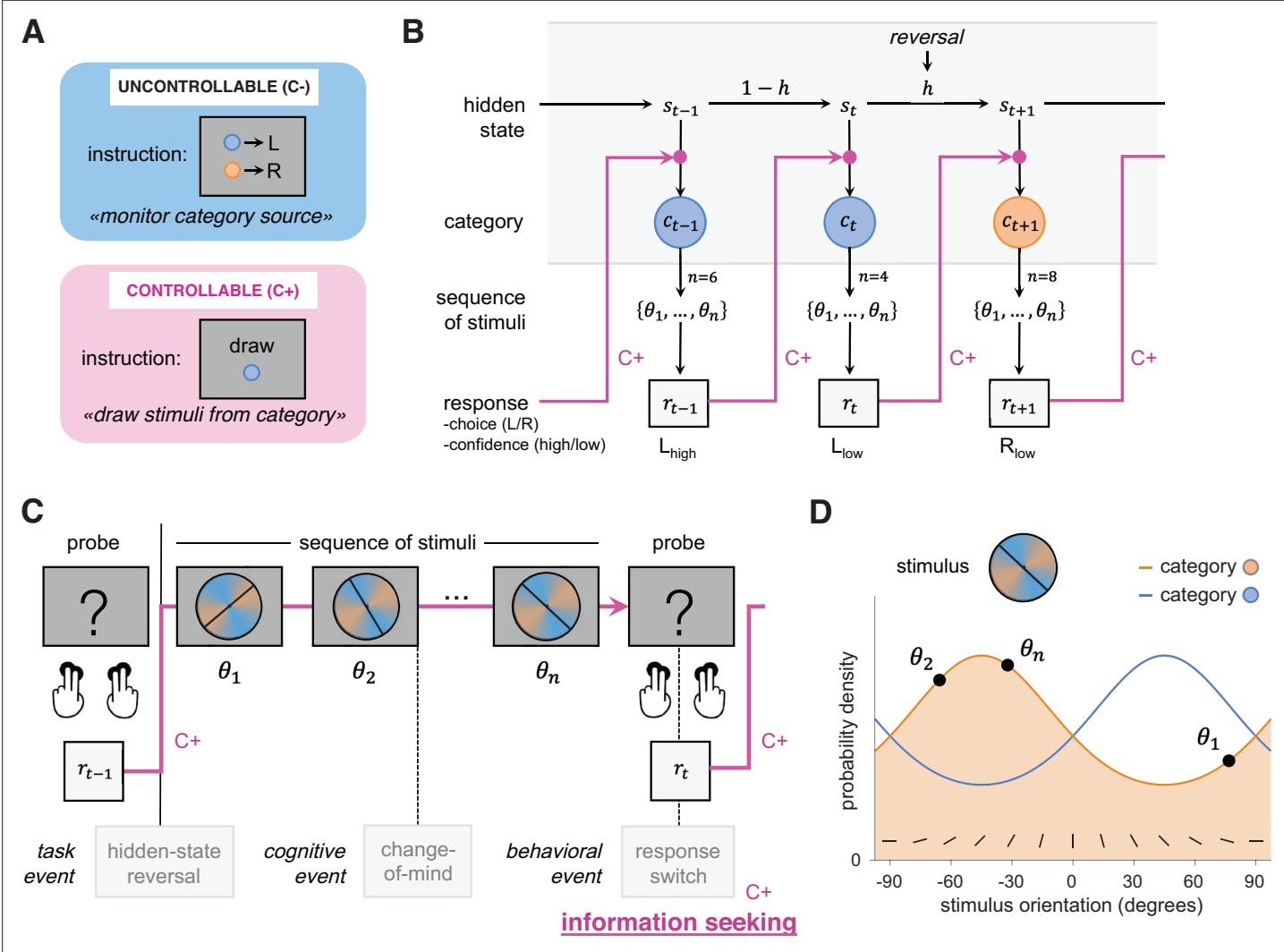

**Figure 1.** Experimental paradigm probing changes-of-mind across two controllability conditions. (**A**) Participants underwent two experimental conditions that varied only in the degree of control over stimuli. In the uncontrollable condition (C-, blue), observers were asked to monitor the category from which stimuli were drawn. In the controllable condition (C+, pink), agents were asked to select an action that will produce stimuli from either category. The evidence available for each choice, but also all stimuli and motor responses were tightly matched across conditions (see 'Materials and methods'). (**B**) Task structure. The hidden state reversed unpredictably after a pseudo-random number of trials. In the C+ condition, the category drawn at trial $t$ only depends on the hidden state $s_t$. In the C- condition, the category drawn at trial $t$ depends on both the hidden state $s_t$ and the previous response $r_{t-1}$, as indicated by pink arrows. (**C**) On each trial, participants were presented with sequences of 2–8 stimuli drawn from either category and were asked to indicate their choice and the associated confidence using four response keys. Confidence keys (low, high) were assigned to the inner and outer keys of each choice (left and right). A schematic sequence of events related to information seeking is depicted. Hidden-state reversals are determined by the experimental design ('task event'). During an exploratory decision, several steps co-occur: a covert change-of-mind ('cognitive event') about the expected reward of the current option; an overt response switch ('behavioral event'); and information seeking, i.e., an active search for information about the new option being considered, which is only possible in the C+ condition, where participants have control. (**D**) Stimuli were oriented bars drawn from either of two categories (orange and blue), whose means are orthogonal to each other. The generative probability distributions of drawn orientation for each category are depicted.

of decision outcomes. Taken together, these features suggest that information seeking increases the saliency of information that is perceived as the direct consequence of one's own actions.

## Results

### Experimental paradigm

To examine the cognitive signatures of information seeking in controllable environments, we asked human participants to perform an adaptive decision-making task consisting of two tightly matched conditions that only vary in the degree of control over information sampling bestowed to participants (*Weiss et al., 2021*). Participants were presented with visual sequences of oriented stimuli drawn from either of two categories (blue or orange) associated with distinct but overlapping probability distributions over orientation (*Figure 1A*; see 'Materials and methods'). In-between each sequence, participants were asked to provide a response regarding the current hidden state of the task, together with a binary confidence estimate in their response (high or low confidence) using four response keys (*Figure 1B*).

In the uncontrollable (C-) condition, participants were asked to monitor the category from which stimuli were drawn and report this category at the end of each sequence. By contrast, in the controllable (C+) condition, the same participants were asked to draw stimuli from either of the two categories. In this C+ condition, the two response keys from one hand were associated with a draw of stimuli from the target category, whereas the other two response keys were associated with a draw from the other (nontarget) category (*Figure 1B*). In the uncontrollable condition, the drawn category reversed unpredictably, requiring participants to adapt their responses to these changes (*Figure 1D*). By contrast, in the controllable condition, it is the association between response keys and categories that reversed unpredictably, also requiring participants to adapt their responses to these changes. In both conditions, participants perform one action per trial. By design, the only difference between conditions is the degree of control bestowed to participants over the sampling of stimuli. Participants were observers monitoring the external source of stimuli in the C- (uncontrollable) condition, whereas participants were agents sampling stimuli through their actions in the C+ (controllable) condition (*Figure 1C*). To experience the difference between experimental conditions, we provide a short gamified analog experiment at infdm.scicog.fr/aodemo/runner.html.

### Psychometric analyses of choice and confidence

By examining differences between conditions, we first established that participants adapted their behavior after a reversal to select the response corresponding to the new hidden state (*Figure 2A*). We found that participants were slower to adapt after a reversal in the C+ condition, with psychometric fits indicating a higher reversal time constant in this controllable condition ($t_{32}$ = 5.0, p=1.95 × $10^{-5}$; see 'Materials and methods'). Participants also reached a higher asymptotic reversal rate in the C+ condition ($t_{32}$ = 6.3, p=4.0 × $10^{-7}$; *Figure 2B*), replicating earlier findings (*Weiss et al., 2021*).

As expected, participants' confidence decreased after a reversal in both conditions (*Figure 2C*). However, confidence decreased more sharply in the C+ (*Figure 2D*), an observation confirmed by psychometric fits of a confidence 'drop' parameter ($t_{32}$ = 3.59, p=0.0011; see 'Methods'). In addition, the confidence time constant characterizing the slope of confidence increase after a reversal was slightly higher in the C+ condition (paired *t*-test, $t_{32}$ = −1.9, p=0.064; Wilcoxon signed-rank test, z = −2.65, p=0.008) (*Figure 2D*). Together, these findings indicate that participants adapted their behavior more slowly and their confidence dropped more strongly in the C+, controllable condition (*Figure 2*).

We then established that participants switched their category choice more often in the C- (32% of trials) than in the C+ (22% of trials) condition ($t_{32}$ = 8.64, p=7.2 × $10^{-10}$). Participants' choice accuracy was slightly higher in the C+ (81.9% correct) than in the C- (78.4% correct) condition ($t_{32}$ = −4.9, p=2.6 × $10^{-5}$). Moreover, response times (RTs) on repeat decisions were similar between C- (mean = 693.2 ms, SEM = 55.4 ms, median = 608.9 ms) and C+ (mean = 677.5 ms, SEM = 45.2 ms, median = 626.7 ms) conditions ($t_{32}$ = −1.52, p=0.14, $BF_{10}$ = 0.531). This was also true in earlier work using the same conditions (see Experiment 4 described below, $t_{23}$ = 0.50, p=0.62, $BF_{10}$ = 0.241) (*Weiss et al., 2021*). These initial results indicate that the differences between conditions are unlikely to be generated by working memory or executive demand differences.

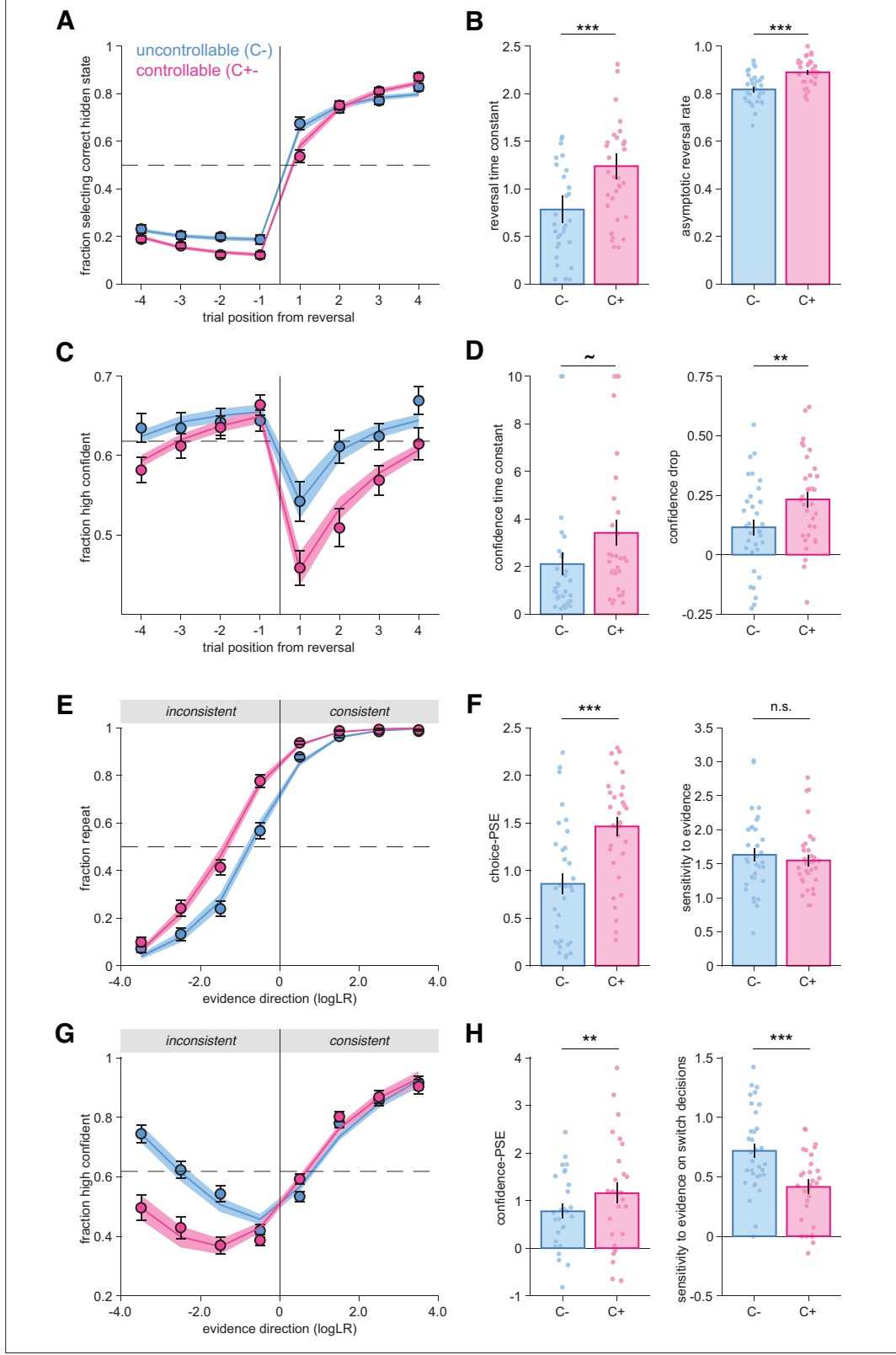

**Figure 2.** Behavioral reversal and repetition curves characterizing participants' responses and their confidence. (**A**) Fraction of hidden state correctly reported as a function of trial number before and after a reversal. Vertical lines indicate the position of reversals. Horizontal dotted line indicates chance level. Circles and error bars indicate mean and SEM across participants (N = 33). Shaded areas indicate mean and SEM of psychometric predictions

*Figure 2 continued on next page*

*Figure 2 continued*

from the best-fitting truncated exponential functions. (**B**) Psychometric parameters 'reversal time constant' and 'asymptotic reversal rate' characterizing the fitted response reversal curve in (**A**) (see 'Materials and methods'). Circles indicate individual participants, and bars and error bars indicate the mean and SEM across participants. \*\*\*p<0.001, paired *t*-tests. (**C**) Fraction of high-confidence responses as a function of trial number before and after a reversal. Horizontal dotted line indicates mean confidence. Circles and error bars indicate mean and SEM across participants (N = 33). Shaded areas indicate mean and SEM of psychometric predictions from the best-fitting sigmoid functions. (**D**) Psychometric parameters 'confidence time constant' and 'confidence drop' for the fitted confidence reversal curve in (**B**). Circles indicate individual participants, and bars and error bars indicate the mean and SEM across participants. \*\*p<0.01, ~p=0.064 paired *t*-test and p<0.008 Wilcoxon signed-rank test. (**E**) Fraction of response repetitions as a function of whether the evidence was consistent (in favor of repeating) or inconsistent with the previous choice. Circles indicate human data (N = 33), and error bars display SEM across participants. Shaded areas indicate mean and SEM of psychometric predictions from the best-fitting truncated exponential functions. (**F**) Psychometric parameters 'point of subjective equivalence' (choice-PSE) and 'sensitivity to evidence' (slope) characterizing the response repetition curves in (**E**). \*\*\*p<0.001, n.s., not significant, paired *t*-tests. (**G**) Fraction of high-confidence responses as a function of whether the evidence was consistent or inconsistent with the previous choice. Circles indicate human data (N = 33), and error bars display within-participant SEM. Within-participant error bars are represented to allow a comparison between conditions without an influence of inter-individual variability about the use and calibration of the high- and low-confidence responses. Shaded areas indicate mean and SEM of psychometric predictions from the best-fitting mixture of two sigmoid functions for repeat and switch trials respectively. (**H**) Psychometric parameters characterizing the confidence repetition curves in (**G**). Left panel: 'confidence-PSE' (point of subjective equivalence) representing the quantity of evidence for which participants are as confident in their repeat decisions as in their switch decisions. \*\*p<0.01, Wilcoxon signed-rank test (see 'Materials and methods'). Right panel: 'sensitivity to evidence' for the switch trials. \*\*\*p<0.001, paired *t*-test. Circles indicate individual participants, and bars and error bars indicate the mean and SEM across participants. In all panels, uncontrollable (C-) condition is in blue and controllable (C+) condition in pink.

The online version of this article includes the following figure supplement(s) for figure 2:

**Figure supplement 1.** Psychometric analysis of choice and confidence reversal and repetition curves in retrospective (purple) and prospective (green) conditions (Experiment 3).

**Figure supplement 2.** Psychometric analysis of Experiment 2B.

**Figure supplement 3.** Behavioral signatures of choice and confidence validate the computational model.

To better understand the origin of the differences between conditions, we reanalyzed participants' choices and confidence as a function of the objective evidence available in favor of repeating their previous choice ('consistent evidence') vs. switching away from their previous choice ('inconsistent evidence'; *Figure 2E–H*). In both conditions, we found that the stronger the consistent evidence, the more often participants repeated their previous choice (*Figure 2E*). Crucially, psychometric fits revealed a difference in the point of subjective equivalence (choice-PSE) between conditions ($t_{32}$ = −7.83, p=6.2 × 10$^{-9}$; *Figure 2F*). The choice-PSE parameter reflects the amount of evidence required for a participant to switch away from their previous choice more often than repeat it. This difference means that participants needed more evidence against their previous decision for switching in the C+ condition. In contrast, participants' sensitivity to the available objective evidence was not significantly different across conditions ($t_{32}$ = 1.09, p=0.28, BF$_{10}$ = 0.323; *Figure 2F*).

We analyzed confidence reports using the same procedure. As expected, confidence increased with the strength of consistent evidence in both conditions (*Figure 2G*). By contrast, participants were considerably less confident in their choices following inconsistent evidence in the C+ condition (*Figure 2G*). To quantify these effects, we fitted two sigmoid functions to switch and repeat choices separately, each sigmoid being characterized by an offset and a slope (see 'Materials and methods'). Based on these fits, we estimated the quantity of evidence for which confidence is equal for repeat and switch decisions (confidence-PSE, *Figure 2G*). We found a significant confidence-PSE difference between conditions (Wilcoxon signed-rank test, z = −2.89, p=0.0038; *Figure 2H*), indicating that participants needed more inconsistent evidence to be equally confident in switch and repeat decisions in the C+ condition. We also found that the sensitivity of confidence reports to evidence (slope) was smaller in the C+ than in the C- condition on switch decisions ($t_{32}$ = 5.1, p=1.6 × 10$^{-5}$; *Figure 2H*).

## Ruling out alternative accounts of differences between conditions

Although the degree of control over evidence sampling is the key difference between conditions, we sought to examine two alternative interpretations. First, we examined whether the temporal direction of inference could explain the observed differences between conditions: prospective in the C+ condition and retrospective in the C- condition (see 'Materials and methods'). Participants were now asked to guess which category the computer *drew* from when they saw the sequence of stimuli (retrospective condition) or guess which category the computer *will draw* from based on what they saw (prospective condition). Neither condition conferred any control to participants. Psychometric analyses of behavior indicate no difference in reversal time constant between retrospective and prospective conditions ($t_{24}$ = −0.016, p=0.98, $BF_{10}$ = 0.21) and a small difference in asymptotic reversal rate ($t_{24}$ = 2.2, p=0.038, $BF_{10}$ = 1.59). Importantly, there was no difference in choice-PSE ($t_{24}$ = −1.67, p=0.11, $BF_{10}$ = 0.70) or sensitivity to evidence ($t_{24}$ = 1.2, p=0.25, $BF_{10}$ = 0.39) between retrospective and prospective conditions (see *Weiss et al., 2021* for a detailed analysis). As expected, participants' confidence decreased after a reversal, but its dynamics were similar across conditions (*Figure 2—figure supplement 1E*). This was revealed in the similar values of the confidence drop parameter across conditions ($t_{24}$ = 0.89, p=0.38, $BF_{10}$ = 0.30) (because confidence drop is not different from zero in both conditions, confidence time constant is not reliably determinable, and therefore are not presented) (*Figure 2—figure supplement 1F*). Furthermore, we found a difference in confidence-PSE ($t_{24}$ = −3.1, p=0.0049), but in contrast to the original conditions of Experiments 1 and 2A, we observed no difference in the sensitivity to evidence parameter on switch decisions ($t_{24}$ = 0.95, p=0.35, $BF_{10}$ = 0.32; *Figure 2—figure supplement 1H*). These results provide evidence that the differences between the original C- and C+ conditions are not due to a retrospective or prospective temporal orientation, but instead indicate differences due to the degree of control over decision evidence.

Second, we examined whether the maintenance of a constant (fixed) category goal across trials was critical for participants to experience a different degree of control between conditions (Experiment 2B). We created a condition in which the instructions (which category to draw from in the C+

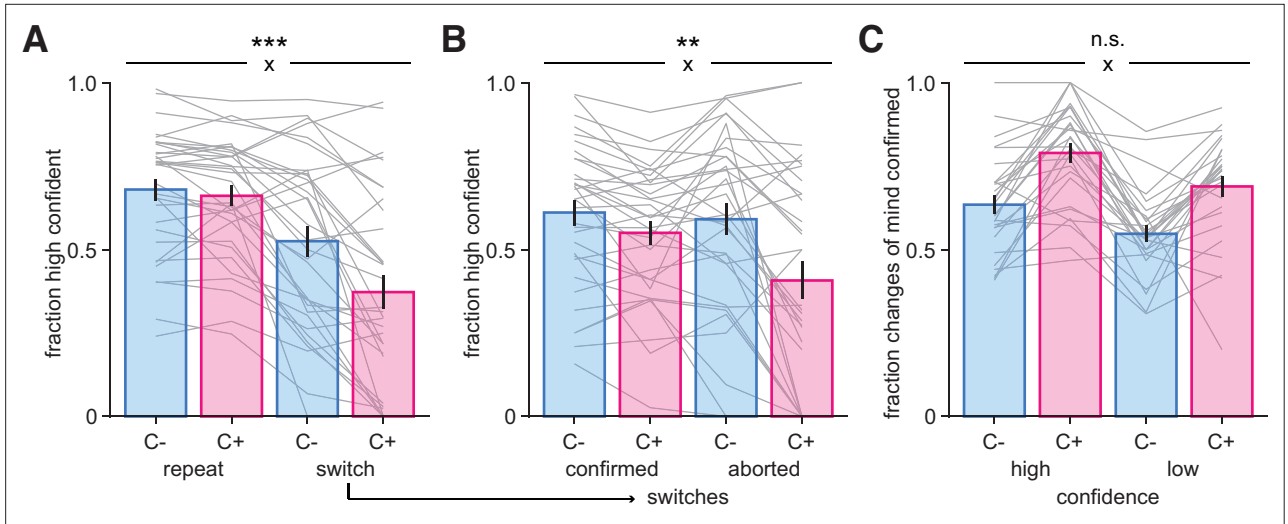

**Figure 3.** A possible role for confidence in changes-of-mind. (**A**) Fraction of high-confidence responses as a function of whether participants repeated their previous choice ('repeat') or changed their mind ('switch') in the uncontrollable (C-, blue) and controllable (C+, pink) conditions. (**B**) Fraction of high-confidence responses following switch decisions that were confirmed or aborted, i.e., when participants return back to their previous response in the two conditions. Bars and error bars indicate mean and SEM across participants, and gray lines display individual data points (N = 33). (**C**) Fraction of switch decisions confirmed on the next trial (see 'Materials and methods') as a function of whether the change-of-mind was performed with high or low confidence, for each condition. Bars and error bars indicate mean and SEM across participants (N = 28, due to some participants not exhibiting all types of responses [see 'Materials and methods']; note that jackknifed statistics with all 33 participants provided virtually identical results). In all panels, statistical significance from an ANOVA is indicated for the interaction (cross) between RESPONSE TYPE (repeat, switch) and CONDITION (C-, C+). ***p<0.001, **p<0.005, n.s., nonsignificant.

The online version of this article includes the following figure supplement(s) for figure 3:

**Figure supplement 1.** Isolating the interaction between controllability and changes-of-mind on confidence.

condition, which category/action mapping to use in the C- condition) changed on a trial basis, instead of on a block basis (see 'Methods'). We found that confidence in trials with inconsistent evidence was less different between conditions (*Figure 2—figure supplement 2*), and the confidence increase after a reversal was similar across conditions (paired *t*-test, $t_{17} = -1.01$, p=0.33; Wilcoxon signed-rank test, $z = -1.02$, p=0.31, $BF_{10} = 0.380$), in line with Experiments 1 and 2B. Moreover, a difference in choice-PSE between conditions ($t_{17} = -6.4$, p=6.4 × 10$^{-6}$) revealed that participants needed more evidence to change their mind in the C+ condition, again in line with Experiments 1 and 2A (*Figure 2—figure supplement 2*). Importantly, even if Experiment 2B effectively changes the number of tasks that need to be performed, we observed a similar choice-PSE between Experiments 2A and 2B (C-: $t_{17} = 0.77$, p=0.45, $BF_{10} = 0.315$; C+: $t_{17} = 0.73$, p=0.47, $BF_{10} = 0.308$), and the difference in choice-PSE between the C- and C+ conditions did not differ between Experiments 2A and 2B ($t_{17} = 0.49$, p=0.63, $BF_{10} = 0.271$).

## Separable effects of confidence and controllability on changes-of-mind

We have seen that participants changed their mind less often in the C+ condition. Following a switch decision, we found that participants were generally less confident when they switched than when they repeated their previous response (main effect of RESPONSE TYPE, $F_{1,32} = 55.8$, p=1.7 × 10$^{-8}$), and were also less confident in the C+ compared to the C- condition (main effect of CONDITION, $F_{1,32} = 17.9$, p=0.00018; *Figure 3A*). We further identified a significant interaction between RESPONSE TYPE and CONDITION ($F_{1,32} = 18.9$, p=0.00014), indicating a more pronounced decrease in confidence on switch decisions in the C+ compared to the C- condition. A logistic regression confirmed these results (see 'Materials and methods'). Repeat trials led to higher confidence than switch decisions ($t_{32} = 7.66$, p=9.8 × 10$^{-9}$), confidence was higher in the C- than in the C+ condition ($t_{32} = 4.05$, p=0.00029), with a significant interaction between RESPONSE TYPE and CONDITION ($t_{32} = -4.43$, p=0.0001), while controlling for evidence level in the same regression model, which positively contributed to confidence, as expected ($t_{32} = 14.69$, p=8.8 × 10$^{-16}$) (*Figure 3—figure supplement 1A*).

After changing their mind, participants could either confirm their category choice on the next trial ('confirmed' switch) or return back to the category selected before the switch occurred ('aborted' switch). We found that after a switch decision participants were less willing to go back to their previous response in the C+ condition, with 72.7% of switches being confirmed, but only 58.9% in the C- condition ($t_{32} = -8.19$, p=2.4 × 10$^{-9}$). This was in the context of a different baseline switch rate differs across conditions, with participants switching choice more often in the C- (32% of trials) than in the C+ (22% of trials) condition. This is consistent with a stronger stickiness tendency observed in the C+ condition, suggesting a reluctance to switch back and forth when participants were in control, and instead a willingness to test again their new category choice. Moreover, participants were more confident on trials in which switch decisions were confirmed compared to aborted ($F_{1,32} = 10.3$, p=0.0030), and more confident in the C- than in the C+ condition ($F_{1,32} = 20.6$, p=0.0001), with an interaction between these factors ($F_{1,32} = 10.7$, p=0.0026), due to participants' confidence decreasing on aborted switches as compared to confirmed switches in the C+ condition (*Figure 3B*). When they are not in control, participants may be more flexible and switch back and forth more easily, as indicated by a similar level of confidence for switches confirmed and aborted in the C- condition. A logistic regression confirmed these results (see 'Materials and methods'), with participants overall being more confident when they confirm than when they abort a change-of-mind ($t_{32} = 5.49$, p=4.6 × 10$^{-6}$), also more confident in the C- than in the C+ condition ($t_{32} = 3.92$, p=0.0004), with a significant interaction between these factors ($t_{32} = -3.38$, p=0.0019), while controlling for evidence level ($t_{32} = 11.9$, p=2.49 × 10$^{-13}$) (*Figure 3—figure supplement 1B*). Moreover, while these results indicate that evidence partly contributes to confidence, as expected, the patterns of evidence strength were markedly different from those of changes-of-mind (compare *Figure 3A and B* to *Figure 3—figure supplement 1C and D*).

Finally, we examined whether switch decisions performed with high (resp. low) confidence more often led to choices being confirmed (resp. aborted) in each condition (*Figure 3C*). We found a main effect of CONFIDENCE on the fraction of switches confirmed ($F_{1,27} = 30.4$, p=7.8 × 10$^{-6}$), meaning that participants confirmed their switch more often when it was made with high confidence. This suggests a causal role for confidence in controlling changes-of-mind, even though we acknowledge that there was no experimentally causal manipulation of confidence here. Switch decisions were more often confirmed in the C+ condition ($F_{1,27} = 47.8$, p=1.98 × 10$^{-7}$), consistent with a higher flexibility of

responses in the C- condition, without an interaction between CONFIDENCE and CONDITION ($F_{1,27}$ = 0.093, p=0.76). Together these findings reveal that participants (1) changed their mind less often in the C+ condition, (2) did so with reduced confidence, and (3) were afterward more willing to confirm a change-of-mind than returning to their previous response compared to the C- condition.

## Computational model with inference noise and metacognitive noise

To further characterize the mechanisms underpinning choice and confidence differences between conditions, we developed a normative Bayesian model (*Figure 4A*). In line with previous work on a closely related task, we endowed it with noisy inference (*Drugowitsch et al., 2016*; see 'Materials and methods'). We observed a similar amount of inference noise between conditions ($t_{32}$ = 1.45, p=0.16), indicating that evidence was integrated equally well across conditions, in line with psycho-metric results (*Figure 2*). We found a lower perceived hazard rate in the C+ relative to the C- condition ($t_{32}$ = 7.46, p=1.7 × 10$^{-8}$), indicating that participants perceived the environment as less volatile when being in control (*Figure 4B*).

To capture the patterns of confidence responses, we further introduced three additional param-eters: confidence threshold, metacognitive noise, and confidence gain associated with switch trials, all capturing unique aspects of confidence response patterns (see 'Materials and methods' and *Figure 4A*). In line with model-free analyses indicating a difference in the fraction of high-confidence responses between conditions, we found a lower confidence threshold in the C- condition, which corresponds to more high-confident responses, compared to the C+ condition ($t_{32}$ = −4.3, p=1.3 × 10$^{-4}$), together with no difference in metacognitive noise ($t_{32}$ = −0.29, p=0.76), or confidence gain for switch trials ($t_{32}$ = 0.10, p=0.92; *Figure 4B*). Using best-fitting parameters fitted to individual partic-ipants, we validated our model by simulating choice and confidence responses, and analyzed the simulated data in the same way as human data (*Palminteri et al., 2017*; *Wilson and Collins, 2019*). We show that simulations provided an accurate fit to participants' choice and confidence responses (*Figure 2—figure supplement 3*). Importantly, we also established that our fitting procedure provided a satisfactory parameter recovery (see 'Materials and methods'). All correlations between generative and recovered parameters were high (all $\rho$ >0.78, all p<10$^{-14}$), while other correlations were low as indicated in a confusion matrix (*Figure 4—figure supplement 1*). Finally, we found no meaningful order effects as to which condition was shown first. Mean accuracy and mean confidence were similar across order groups (independent $t$-tests, all $t_{31}$ < −1.2, all p>0.23). Likewise, there were no order effects for any of the best-fitting parameters (for all comparisons between conditions, all $t_{31}$ < 1.85, all p>0.074).

We further validated the independent role of each parameter in two ways. First, we examined correlations between best-fitting parameters across participants (*Figure 4—figure supplement 1*). We found a significant negative correlation between inference noise and hazard rate ($\rho$ = −0.51, p=0.0025), in line with a previously reported trade-off between these two sources of variability (*Weiss et al., 2021*). We also found a borderline correlation between confidence threshold and hazard rate ($\rho$ = −0.36, p=0.0409). However, all other correlations were not significant (all $\rho$ < 0.31, all p>0.074), indicating that each parameter captured independent portions of the variance (*Figure 4—figure supplement 1*). Second, we did a median split of participants into groups of high and low param-eter values (*Figure 4—figure supplements 3 and 4*), each parameter having a selective influence on choices and confidence. Even when parameters were similar across conditions (e.g., confidence gain), there was still a substantial inter-individual variability that had a visible effect on participants' confidence, indicating the necessity of each parameter in capturing qualitative signatures participants' choices and confidence.

We next examined whether our computational model could not only predict choice and confi-dence (*Figure 2—figure supplement 3*), but also predict effects on the next decision. We refitted the psychometric choice-PSE (*Figure 2F*) separately for subsamples of trials for both human data and simulated choice data from the best-fitting parameters (see 'Materials and methods'). For human choices in the C- condition, we found a higher PSE following a repeat compared to a switch trial, indicating that more evidence was required to switch away from the current best option ($t_{32}$ = 3.2, p=0.0033), whereas in the C+ condition, PSEs were similar after repeat and switch trials ($t_{32}$ = 0.08, p=0.93; *Figure 5A*, bars). In the C- condition, the PSE pattern predicted by the model matched human choices ($t_{32}$ = 5.4, p=7.0 × 10$^{-6}$), with a smaller PSE after a switch than after a repeat trial (*Figure 5A*,

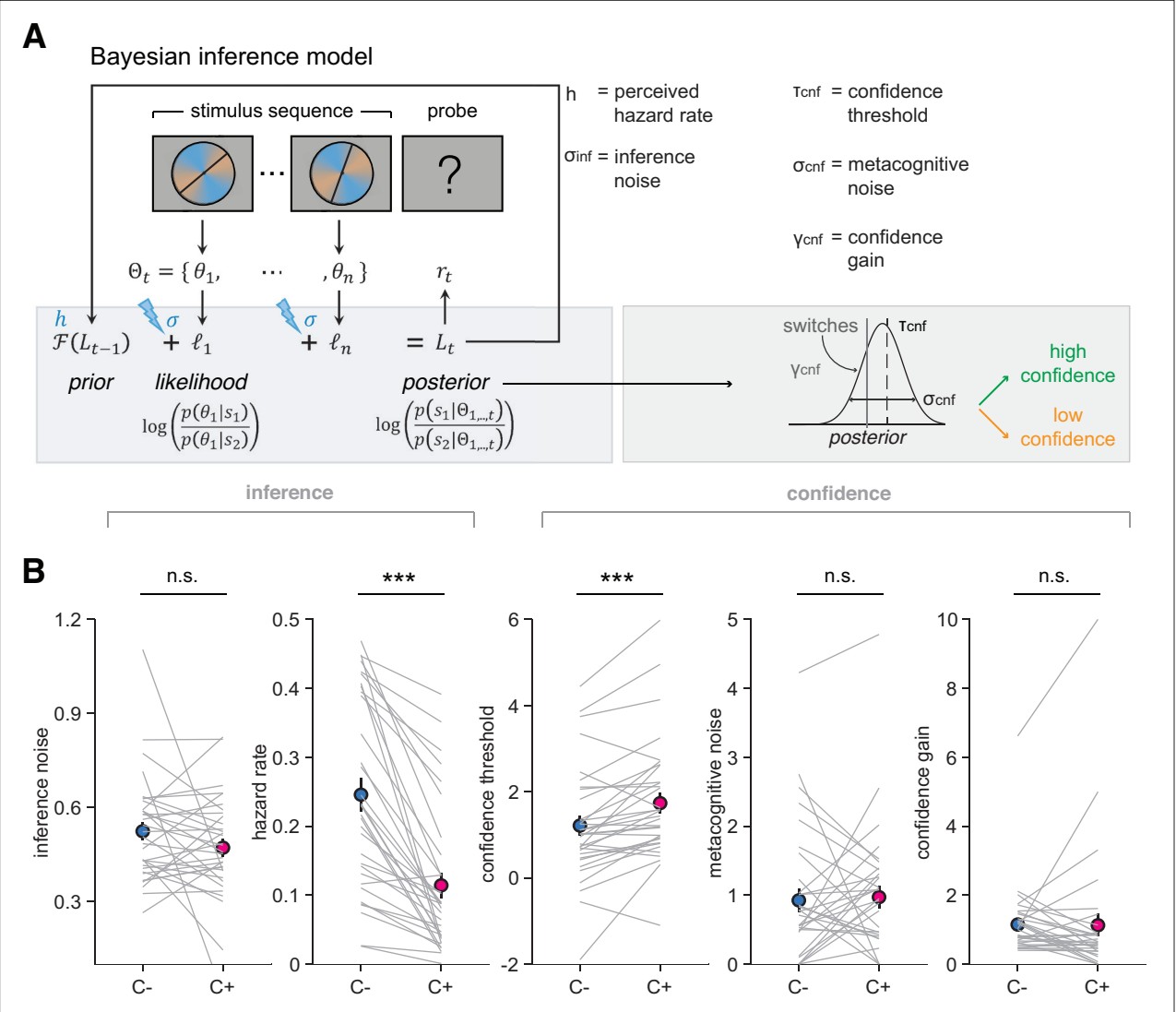

**Figure 4.** Bayesian normative model describing choices and confidence. (**A**) Model schematic of computations. The model updates a belief about category from the evidence acquired through the sequence of stimuli, additionally corrupted by inference noise that scales with sequence length, and with a hazard rate controlling reversal occurrence. Based on the strength of the belief about category, responses above (resp. below) a confidence threshold are given with high (resp. low) confidence, with choices and confidence therefore being based on the same posterior belief. An imperfect readout of the posterior belief is modeled by metacognitive noise, and a confidence gain parameter selectively applied of switch trials (see 'Materials and methods'). (**B**) Best-fitting model parameters: inference noise, hazard rate, confidence threshold, metacognitive noise, and confidence gain in the uncontrollable (C-, blue) and controllable (C+, pink) conditions. Circles and error bars indicate mean and SEM across participants (N = 33), and gray lines display individual parameter values. \*\*\*p<0.00001, n.s., nonsignificant, paired *t*-tests.

The online version of this article includes the following figure supplement(s) for figure 4:

**Figure supplement 1.** Parameter recovery analysis.

**Figure supplement 2.** Correlation between model parameters.

**Figure supplement 3.** Effect of inference noise and hazard rate parameters on choice and confidence patterns.

**Figure supplement 4.** Effect of confidence threshold, metacognitive noise, and confidence gain parameters on confidence patterns.

squares). By contrast, in the C+ condition, the PSE pattern predicted by the model showed the same difference between repeat and switch trials ($t_{32}$ = 8.9, p=3.3 × 10$^{-10}$), unlike what was observed for human choices (interaction between DATA TYPE [human, model] and RESPONSE TYPE [after a repeat, after a switch]; $F_{1,32}$=16.94, p=0.000025; *Figure 5A*). This deviation from model predictions indicates that

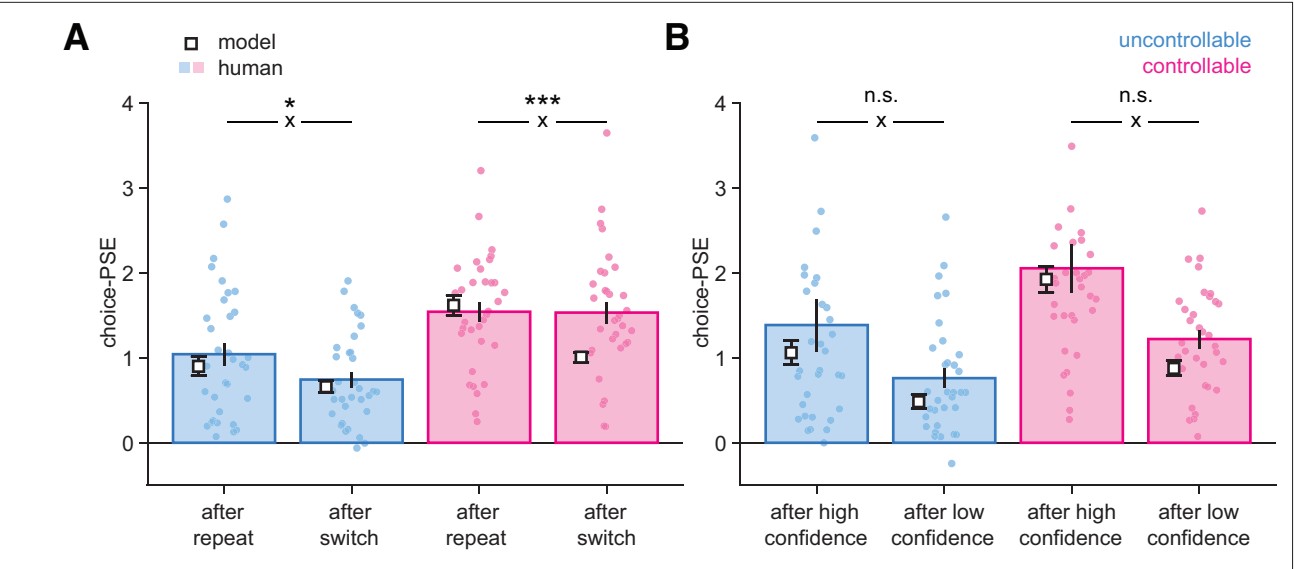

**Figure 5.** Bayesian normative model describing choices and confidence. (**A**) Psychometric parameter choice-PSE (point of subjective equivalence) fitted separately on trials following a repeat vs. a switch in the uncontrollable (C-, blue) and controllable (C+, pink) conditions. Statistical significance from an ANOVA is indicated for the interaction (cross) between RESPONSE TYPE (after a repeat, after a switch) and DATA TYPE (human, model). *p=0.027, ***p=0.000025, n.s., nonsignificant. (**B**) Psychometric parameter choice-PSE fitted separately on trials following a high-confidence vs. a low-confidence response in each condition. Small circles indicate individual data points (N = 33), bars and error bars indicate mean and SEM for human data. White squares and error bars indicate mean and SEM for model simulations of different participant sessions. Statistical significance from an ANOVA is indicated for the interaction (cross) between RESPONSE TYPE (after high confidence, after low confidence) and DATA TYPE (human, model). n.s., nonsignificant.

participants required more evidence to switch back following a change-of-mind in the controllable C+ condition.

We further investigated whether confidence had a role in this deviation. We again fitted choice-PSEs separately for trials following a high- or a low-confidence response in each condition (***Figure 5B***). As expected, PSEs were lower following a low-confidence response, indicating that participants are more willing to change their minds after a low-confident response. Importantly, and unlike what was observed when comparing responses following repeat and switch trials, we observed a similar qualitative pattern for human and model choices (***Figure 5***). This indicates that the difference in PSE following switches in the C+ condition cannot be due to the model having miscalibrated confidence because the differences in PSE observed for human choices across confidence levels and between conditions were adequately captured (***Figure 5B***, ***Figure 2—figure supplement 3***).

## Neuromagnetic signatures of information seeking during changes-of-mind

We examined whether the aforementioned features of switch decisions in the C+ condition were related to changes in cortical, subcortical, and peripheral systems. First, we analyzed fluctuations in spectral power of brain signals recorded in MEG in the alpha band (Experiment 4), a frequency range known to be associated with changes in attentional and arousal systems (***Corbetta and Shulman, 2002***).

We computed a 'repeat minus switch' contrast in the alpha-band power at [−4, + 4] s around the response probe onset in all channels. Then, we clustered the channels according to the strength of pairwise correlations of time courses of this contrast across channels (***Figure 6A***). This analysis identified two main spatial clusters of channels: an occipital cluster and a frontal cluster. Inspecting time courses of the 'repeat minus switch' contrast (hereafter, 'switch effect'; ***Figure 6B***) revealed that the switch effect occurred earlier in the occipital cluster (time from trial onset, occipital: 1.450 ± 0.229 s, frontal: 2.099 ± 0.243 s, jackknifed mean ± SEM; jackknifed $t_{23}$ = 3.1, p=0.005) and peaked before the response probe onset (time from probe onset, occipital: −0.393 ± 0.028 s; jackknifed $t_{23}$ = −14.2, p<0.001). In contrast, the switch effect started later and peaked after the probe onset (+0.958 ± 0.032 s; jackknifed $t_{23}$ = 29.6, p<0.001) in the frontal cluster (***Figure 6B***). The two clusters were

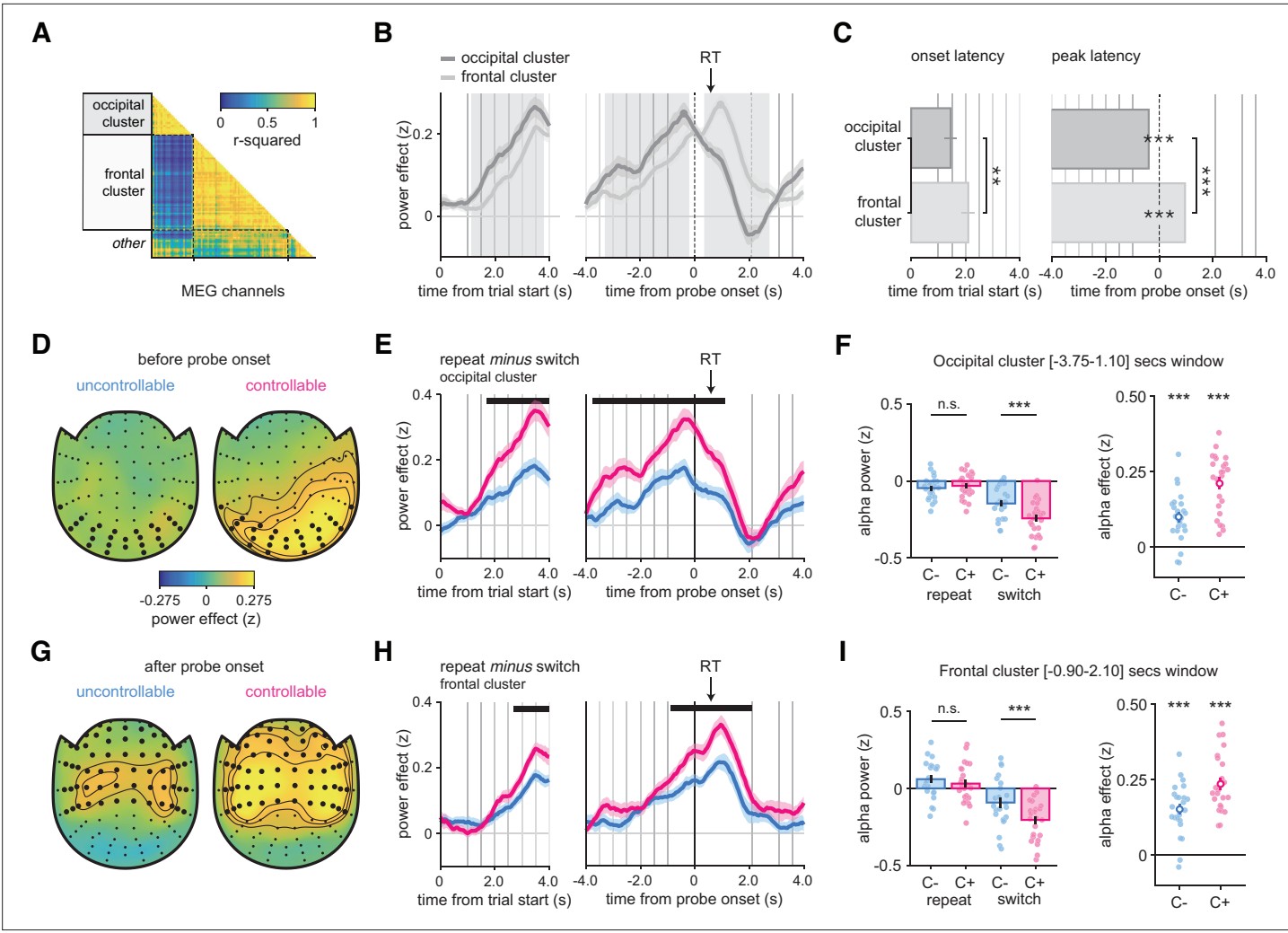

**Figure 6.** Alpha-band magnetoencephalography (MEG) as a window onto the neural basis of changes-of-mind (Experiment 4). (**A**) Pairwise correlation matrix of time courses of alpha-band contrast between repeat and switch trials across all MEG channels, grouped into two main clusters, labeled as occipital and frontal based on their localization. (**B**) Average time course of alpha-band contrast between repeat and switch trials for the occipital (dark gray) and frontal (light gray) clusters, time-locked at trial onset (left panels) or response probe onset (right panels). Shaded areas indicate time windows (1: before probe onset, 2: after probe onset) in which time courses in the two clusters significantly differ. (**C**) Left panel: onset latency of the contrast between repeat and switch trials, locked to trial onset. Right panel: peak latency of the contrast between repeat and switch trials, time-locked to response probe onset. Bars and error bars indicate jackknifed means and SEM. (**D, G**) Spatial topography of the contrast between repeat and switch trials in the time window (1) before probe onset (**D**) and in the time window (2) after probe onset (**G**), in the uncontrollable (left) and controllable (right) conditions. (**E, H**) Contrast of repeat minus switch trials in the C- (blue) and C+ (pink) conditions time-locked at the trial onset (left panels) or response probe onset (right panels). Horizontal lines indicate statistical significance at p<0.05 corrected. Vertical lines indicate trial events: probe onset, samples of the sequence, and the response probe onset. The black arrow indicates the average response time across conditions and participants. (**F, I**) Left panels: stars indicate significance from post hoc tests from an ANOVA with CONDITION (C-, C+) and RESPONSE TYPE (repeat, switch) as within-participant factors on the time window statistically significant after correction identified in (**E**) and (**H**), respectively. Bars and error bars indicate mean and SEM. Right panels: stars indicate significance for paired *t*-tests of the contrast between repeat and switch trials in each condition separately. Circle and error bar indicate mean and SEM. *p<0.05, **p<0.01, ***p<0.001. N = 24 participants. See also *Figure 6—figure supplement 1* for raw effects in occipital and frontal clusters.

The online version of this article includes the following figure supplement(s) for figure 6:

**Figure supplement 1.** Raw time courses of magnetoencephalography (MEG), pupil, and interbeat interval (IBI) data.

**Figure supplement 2.** Motor response preparation effects in alpha-band magnetoencephalography (MEG) (Experiment 4).

**Figure supplement 3.** Physiological analyses controlling for sequence length.

therefore defined by two distinct time windows. The occipital switch effect was larger than the frontal effect before the response probe onset (during the sequence, peak $t_{23}$ = 6.0, cluster-level p<0.001), whereas the frontal effect was larger than the occipital effect after the probe onset (during and after the response, peak $t_{23}$ = 7.5, cluster-level p<0.001; *Figure 6C*).

In the occipital cluster (*Figure 6D*), the switch effect was larger in the C+ than the C- condition (peak $t_{23}$ = 6.2, cluster-level p<0.001), with the largest difference just before probe onset (*Figure 6E*, see also *Figure 6—figure supplement 1A and B* for the raw effects). We found no significant difference between conditions on repeat trials ($t_{23}$ = 0.6, p=0.528) but a significant one on switch trials ($t_{23}$ = 3.9, p<0.001) (*Figure 6F*) (interaction: $F_{1,23}$ = 40.7, p<0.001; *Figure 6E*). The frontal cluster (*Figure 6G*) featured a larger (and longer) switch effect in the C+ than the C- condition (peak $t_{23}$ = 4.4, cluster-level p<0.001), with the largest difference just after probe onset (*Figure 6H*, see also *Figure 6—figure supplement 1C and D* for the raw effects). As for the occipital cluster, we found no significant difference between conditions on repeat trials ($t_{23}$ = 1.1, p=0.27), but a significant one on switch trials ($t_{23}$ = 4.1, p<.001) (*Figure 6I*) (interaction: $F_{1,23}$ = 24.6, p<0.001; *Figure 6H*). Thus, in both clusters, switch decisions are driving the differences between conditions. Since previous work has identified links between motor preparation and alpha-band activity, we further sought to examine whether these switch effects were due to response preparation. For this purpose, we selected a subsample of channels sensitive to response-hand selectivity in the frontal cluster. We found no difference in motor lateralization between conditions at a liberal sample-wise threshold p<0.05. This suggests that the effects described above are truly about changes-of-mind, not the preparation of a response switch (*Figure 6—figure supplement 2*).

We reasoned that if participants treated all trials independently, repeat or switch decisions would have a similar 'status' and we should observe a choice-PSE close to zero. This should be related to a lower alpha suppression (repeat minus switch effect) because repeat and switch trials are treated almost similarly, irrespective of the condition. Therefore, for each condition we examined whether a lower alpha suppression effect (*Figure 6F* for the occipital cluster and *Figure 6I* for the frontal cluster) was associated with a lower choice-PSE (i.e., closer to zero). In the occipital cluster, we found significant correlations between the alpha effect and the choice-PSE across participants in the C- ($\rho$ = 0.496, p=0.013) and C+ ($\rho$ = 0.585, p=0.002) condition. In the frontal cluster, however, the direction of correlations was similar, but these effects were not statistically significant (C- condition: $\rho$ = 0.311, p=0.137; C+ condition: $\rho$ = 0.357, p=0.086). Likewise, in the occipital cluster, we observed significant correlations between the alpha effect and the best-fitting hazard rate (the model-based counterpart of choice-PSE) across participants in the C- ($\rho$ = −0.467, p=0.021) and C+ ($\rho$ = −0.605, p=0.001) conditions. In the frontal cluster, however, the correlations were similar though not statistically significant (C- condition: $\rho$ = −0.345, p=0.098; C+ condition: $\rho$ = −0.347, p=0.096). These correlations should be taken with caution since our sample size is relatively small and our experiments were not designed and powered to examine them.

## Pupillometric and cardiac signatures of information seeking during changes-of-mind

We analyzed pupil dilation time-locked at the trial onset and response probe onset (pooled over Experiments 1 and 2A; see 'Materials and methods'). In both conditions, pupil dilation increased slightly before, but mainly after probe onset, with a first peak around 1 s after probe onset and a second peak corresponding to the next trial (see *Figure 6—figure supplement 1E and F* for raw effects). To compare pupil dilation between switch and repeat trials, we compared the time courses of these two trial types in *Figure 7A*, and, at each time point, performed a 2 × 2 ANOVA with RESPONSE TYPE (repeat, switch) and CONDITION (C-, C+) as within-participant factors. This revealed a large time window (time from probe onset, +0.93 to +7.77 s), in which the interaction between RESPONSE TYPE and CONDITION was significant ($p_{corr}$<0.001; *Figure 7B*). Within this time window, pupil dilation was similar across conditions on repeat trials ($t_{30}$ = 1.37, p=0.18), but was larger in the C+ than in the C- condition on switch trials ($t_{30}$ = −5.48, p=5.94 × $10^{-6}$). At the trial onset, we also found a time window with a significant interaction between CONDITION and RESPONSE TYPE (starting at +3.45 s from probe onset, $p_{corr}$<0.001; *Figure 7A*).

To investigate whether the peripheral nervous system also contains specific signatures of information seeking, we examined the cardiac interbeat interval (IBI) from electrocardiogram (ECG)

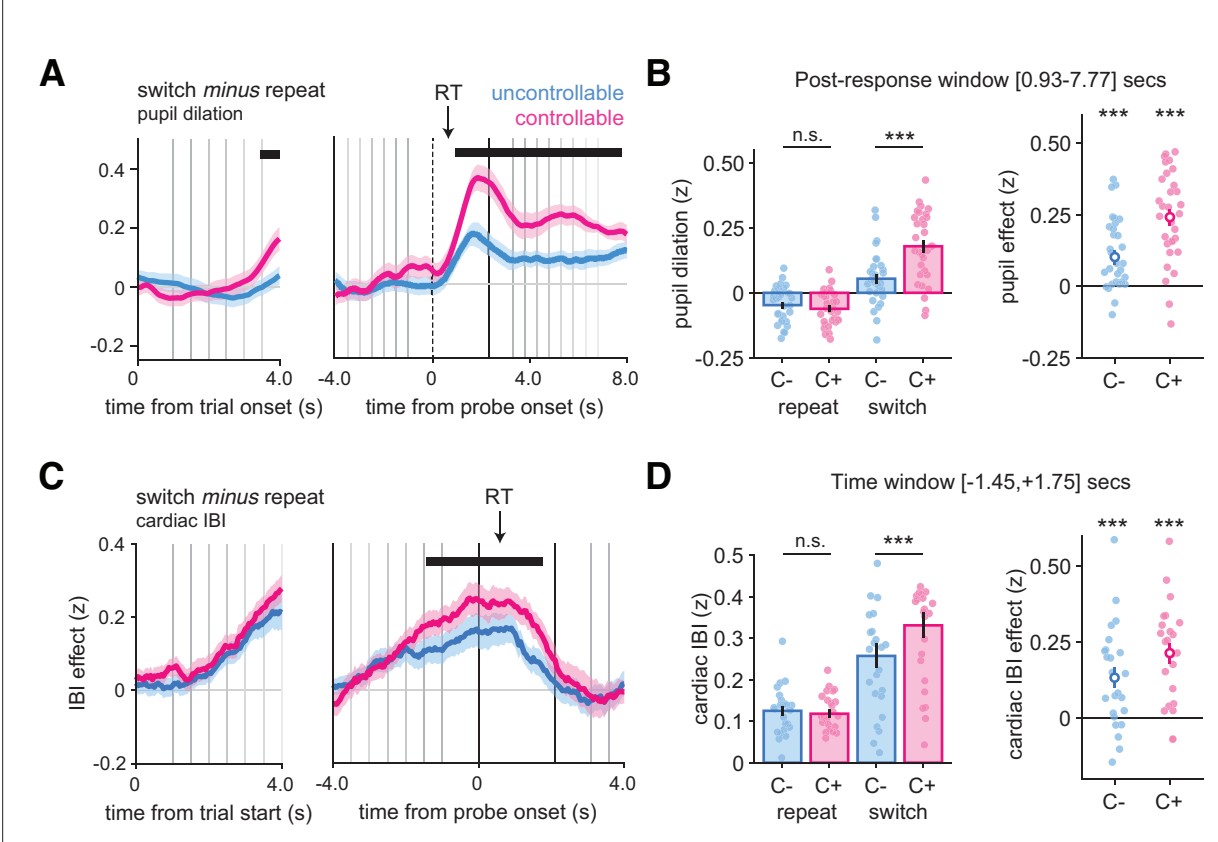

**Figure 7.** Larger pupil dilation and cardiac interbeat interval (IBI) signals on changes-of-mind. (**A**) Pupil dilation on switch minus repeat decisions time-locked at the trial onset (left panels) and response probe onset (right panels). Shaded areas indicate mean and standard error across participants at each time point. The horizontal lines indicate statistical significance at p<0.05 corrected from ANOVAs with CONDITION (C-, C+) and RESPONSE TYPE (repeat, switch) as within-participant factors. The black arrow indicates the average response time across conditions and participants (N = 31, pooled across Experiments 1 and 2). (**B**) Post hoc ANOVA effects on the time windows statistically significant after correction using permutation tests identified in (**A**). Stars correspond to significance of paired *t*-tests between conditions in the time window, ***p<0.001, n.s., not significant. Bars and error bars indicate mean and SEM (N = 31). (**C**) Cardiac IBI on switch minus repeat decisions time-locked at the trial onset (left panels) and response probe onset (right panels). Shaded areas indicate mean and standard error across participants at each time point (N = 24, Experiment 4). (**D**) Post hoc ANOVA effects on the time window statistically significant after correction using permutation tests identified in (**C**). Stars indicate significance for paired *t*-tests of the contrast between switch and repeat trials in each condition separately (***p<0.001). Circle and error bar indicate mean and SEM (N = 24). In all panels, vertical lines indicate trial events: trial onset, samples of the evidence, and the response probe onset (dotted line). Blue: C- condition; pink: C+ condition. See also *Figure 6—figure supplement 1* for raw dynamics of pupil response and cardiac IBI.

signals recorded in Experiment 4. In both conditions, the IBI increased before probe onset and slowly decreased thereafter (see *Figure 6—figure supplement 1G and H* for raw effects), a dynamic comparable to that of the alpha-band response suppression observed in MEG. This increase was larger on switch than on repeat trials (C-: $t_{23}$ = 3.8, p<0.001; C+: $t_{23}$ = 6.5, p<0.001; *Figure 7C*), an effect that was more pronounced in the C+ than in the C- condition (main effect of CONDITION: $F_{1,23}$ = 12.6, p=0.002; main effect of RESPONSE TYPE: $F_{1,23}$ = 34.8, p<0.001; interaction RESPONSE TYPE and CONDITION: $F_{1,23}$ = 5.8, p=0.024; *Figure 7D*).

We further examined whether these physiological markers of changes-of-mind can indicate when commitments to a change-of-mind occur: whether they accompany the stimulus sequence progression, already ramping up before the response; or whether they occur when the response is provided. We separated physiological data for short (2–4) and long (6–8) stimulus sequences (see 'Materials and methods'). We observed that the alpha-band effects associated with switch responses arose earlier for long compared to short sequences across conditions, both in the occipital (*Figure 6—figure supplement 3A*) and frontal (*Figure 6—figure supplement 3B*) clusters (main effect of SEQUENCE LENGTH, occipital: peak $F_{1,23}$ = 25.1, cluster-level p<0.001; frontal: peak $F_{1,23}$ = 17.7, cluster-level p<0.001). Thus, the difference between conditions observed for changes-of-mind starts before probe onset,

with participants engaging more attention early in sequences that eventually lead to a response switch. In contrast, pupil dilation did not differ significantly across the two sequence length groups. In both conditions, pupil dilation on changes-of-mind increased only after the response (*Figure 6—figure supplement 3C*). Finally, the IBI revealed dynamics comparable to those of the alpha-band suppression, with differences between short and long sequences restricted to before the probe onset (main effect of SEQUENCE LENGTH: peak $F_{1,23}$ = 10.4, cluster-level p<0.05; *Figure 6—figure supplement 3D*). Together, these results highlight distinct within-trial dynamics, with alpha-band power and IBI effects of changes-of-mind already starting pre-switch, whereas pupillary effects of changes-of-mind being restricted to post-switch.

## Isolating the effects of information seeking from model variables

Since changes-of-mind occurred (1) more often when decision evidence was inconsistent with the previous choice (*Figure 3*) and (2) less often when participants' prior beliefs were stronger, we reasoned that these two variables may differently influence changes-of-mind in each condition, which would impact a pure effect of changes-of-mind. In other words, if decision evidence and prior belief differ across repeat and switch trials, fluctuations in these two quantities may differently contribute to the neurophysiological time courses observed. We thus sought to examine whether neurophysiological effects in the C+ condition were specifically related to changes-of-mind, over and above (1) the absolute strength of prior belief and/or (2) the amount of decision evidence signed by the prior belief (hereafter, 'prior belief' and 'evidence direction'). To account for these two quantities, we relied on the variables inferred from our computational model (see 'Materials and methods').

First, using particle filtering, we extracted trajectories of these two quantities conditioned on participants' responses and established the statistical relationships between these quantities and changes-of-mind. As expected, we found that absolute prior beliefs were stronger on a repeat than on a switch trial ($F_{1,32}$ = 59.02, p=9.3 × 10⁻⁹) and higher than in the C+ than in the C- condition ($F_{1,32}$ = 72.3, p=1.0 × 10⁻⁹), in line with a lower perceived hazard rate found in the C+ condition, with an interaction between RESPONSE TYPE and CONDITION ($F_{1,32}$ = 11.7, p=0.0017). We also established that evidence direction strongly was positive (resp. negative) on repeat (resp. switch) decisions ($F_{1,32}$ = 7527.9, p<1 × 10⁻¹¹), an effect that was more pronounced in the C+ than in the C- condition ($F_{1,32}$ = 52.5, p=3.1 × 10⁻⁸) with an interaction between RESPONSE TYPE and CONDITION ($F_{1,32}$ = 5.59, p=0.024).

In both conditions, both prior belief and evidence direction showed strong and sustained correlations with alpha power suppression in occipital and frontal clusters (*Figure 8A*). In the occipital cluster, when controlling for the effects of both model variables, the C- switch effect disappears before probe onset, but remains afterward. By contrast, the C+ switch effect remained present, albeit smaller, before and after probe onset (residual variance, *Figure 8B*). Importantly, the difference in switch effect remained significant when accounting for these two model variables (interaction RESPONSE TYPE and CONDITION: $F_{1,23}$ = 21.9, p<0.001), meaning that the original difference between conditions (*Figure 6*) cannot be explained away by different prior beliefs or decision evidence. In the frontal cluster, all effects again fully remained after controlling for prior belief and decision evidence (interaction RESPONSE TYPE and CONDITION: $F_{1,23}$ = 18.0, p<0.001; residual variance, *Figure 8B*).

Likewise, in pupil dilation, when controlling for the effects of both variables, the switch effect remained larger in the C+ than in the C- condition (interaction RESPONSE TYPE and CONDITION: $F_{1,30}$ = 8.9, p=0.005) (*Figure 8—figure supplement 1*). Like alpha power suppression, prior belief also showed a sustained positive correlation with cardiac IBI, and evidence direction was negatively associated with cardiac IBI (*Figure 8—figure supplement 1*). When controlling for both variables, the switch effect remained in both conditions, albeit smaller, but the difference between conditions was no longer significant, indicating that evidence direction was driving the switch effect observed in cardiac IBI (compare *Figure 7C* and *Figure 8—figure supplement 1*) (interaction RESPONSE TYPE and CONDITION: $F_{1,23}$ = 0.3, p=0.57). Finally, additionally controlling for the effect of RTs, together with prior belief and evidence direction, provided virtually identical results for all physiological modalities.

## Discussion

Accurate decisions in uncertain environments require forming and updating beliefs through efficient information sampling. Previous studies of human decisions and confidence have often relied on tasks

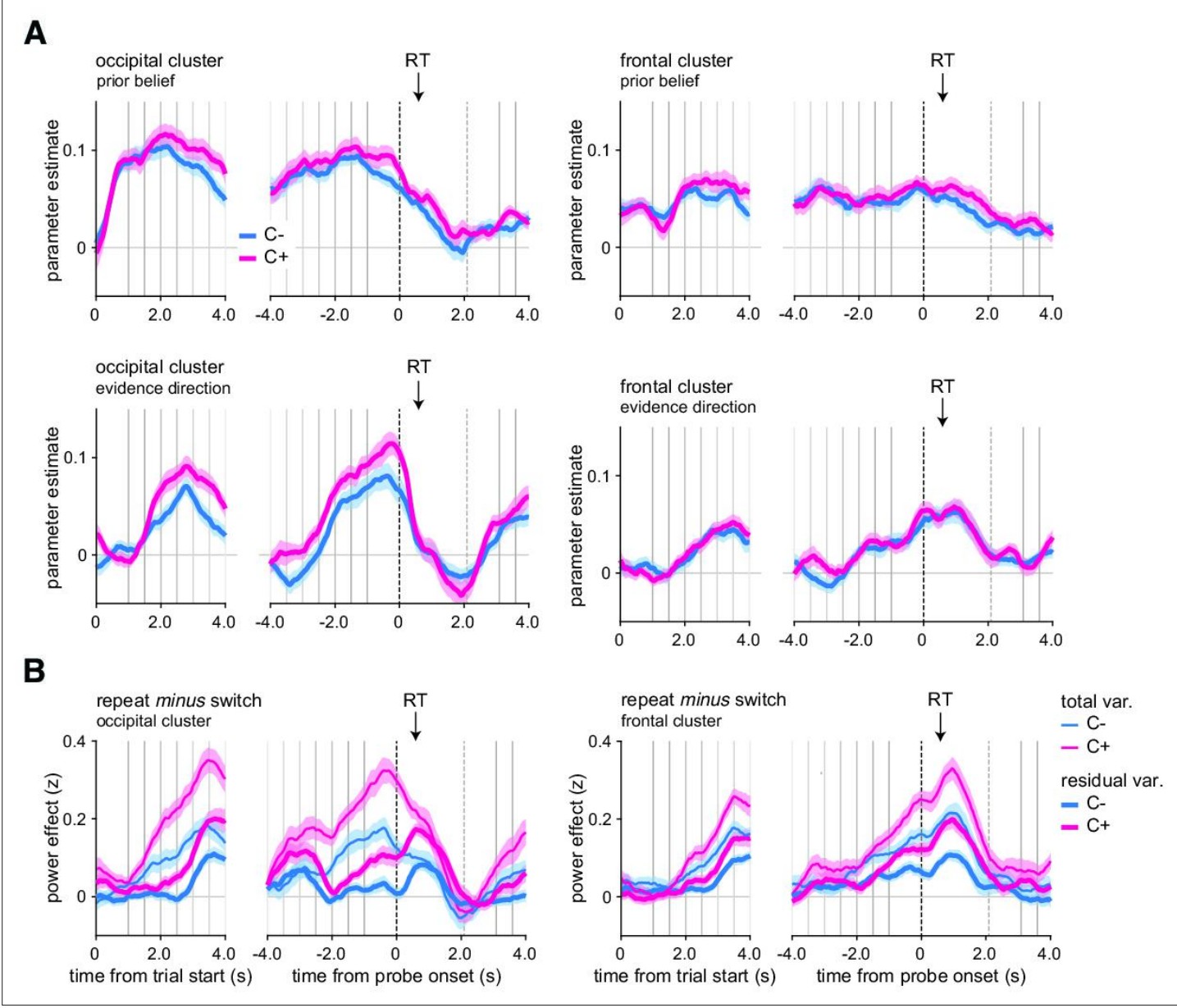

**Figure 8.** In controllable environments, changes-of-mind effects cannot be explained away by fluctuations in prior beliefs or evidence direction.
(**A**) Effects of prior belief and evidence direction on magnetoencephalography (MEG) alpha-band power in the occipital cluster (left panels) and frontal cluster (right panels). (**B**) For MEG alpha-band power in the occipital cluster (left panels) and frontal cluster (right panels), contrast between repeat and switch trials time-locked at the trial onset (left panels) or response probe onset (right panels) after removing the variance due to fluctuations in prior belief and evidence direction (thick lines, residual variance) in the uncontrollable (blue, C-) and controllable (pink, C+) conditions. Raw contrasts are displayed for comparison (thin lines, total variance). Vertical lines indicate trial events: probe onset, samples of the sequence, and response probe onset (dotted line). The black arrow indicates the average response time across conditions and participants. total var.: total variance; residual var.: residual variance after accounting for the two model variables, prior belief and evidence direction. Shaded error bars indicate SEM (N = 24).

The online version of this article includes the following figure supplement(s) for figure 8:

**Figure supplement 1.** Changes-of-mind effects after accounting for fluctuations in prior beliefs or evidence direction.

in which participants have no control over the sampling of information (*Fleming et al., 2018*; *Murphy et al., 2016*; *van den Berg et al., 2016*). However, outside the laboratory, we usually have some control over information sampling: which way to look, which food to try, which person to listen to. We do not passively *receive* information but rather actively *seek* information for decision-making (*Gureckis and Markant, 2012*; *Sharot and Sunstein, 2020*). Here, we reasoned that control is

necessary for information seeking, which we used to dissociate information seeking from changes in beliefs and behavior. Across several variants of the same task tested with and without control over information sampling, we obtained converging evidence that information seeking is associated with decreased confidence and drives active hypothesis testing. At the physiological level, information seeking is associated with stronger and longer-lasting correlates of attention and arousal.

In previous work, information seeking has often been studied and theorized in the context of 'exploration-exploitation' dilemmas (*Costa et al., 2019*; *Daw et al., 2006*). In these tasks, participants are typically immersed in controllable environments and asked to sample one among several options to maximize reward. Therefore, they have to choose between exploiting a currently rewarding option and foregoing rewards to explore alternative options to seek information about their associated rewards (*Rich and Gureckis, 2018*; *Wilson et al., 2014*). This cognitive trade-off raises the issue that exploration and exploitation not only differ in terms of information seeking, but also in terms of other cognitive and behavioral events. In particular, three main events co-occur during an exploratory decision: (1) a covert change-of-mind about the expected reward of the current option, (2) an overt change in behavior (e.g., a response switch), and (3) information seeking, that is, an active search for information about the new option being considered. The latter is only possible in the controllable condition, allowing us to isolate information seeking by comparing response switches between the two conditions.

We fitted a Bayesian inference model to participants' choices and extended our model to predict confidence reports. Simulations with best-fitting parameters of individual participants confirmed that each parameter captures a specific behavioral gradient matching participants' psychometric signatures (*Figure 4—figure supplements 3 and 4*). First, we replicate our previous finding of a lower perceived hazard rate in the controllable condition (*Figure 5B*), corresponding to participants perceiving contingencies as more stable in controllable environments (*Weiss et al., 2021*). This lower perceived hazard rate contributes to explaining a larger loss of confidence on changes-of-mind in the controllable condition. In the model, the lower perceived hazard rate creates stronger prior beliefs, such that more inconsistent evidence is required for them to be reversed. This mechanism predicts lower confidence during changes-of-mind despite stronger evidence in favor of a change-of-mind in the current trial. This increase in the magnitude of prior beliefs also explains why participants confirm more often their changes-of-mind in the controllable condition, despite lower confidence in these changes-of-mind.

In both conditions, our model predicts that the quantity of inconsistent evidence required for participants to change their minds (reflected in choice-PSE estimates) should be smaller following a switch than a repeat decision (*Figure 5A*). This prediction was verified in the uncontrollable condition. By contrast, in the controllable condition, participants violated this prediction by showing similar choice-PSEs following repeat and switch decisions (*Figure 5A*). This deviation between model and data in this condition where changes-of-mind are associated with information seeking could reflect at least two cognitive effects. First, participants may believe a new reversal in task contingencies to be less likely just after a perceived reversal. Second, participants may engage in an active form of hypothesis testing; that is, testing that the new action draws from the target category. Such hypothesis testing is by definition only possible in the controllable condition, where participants can actively sample the environment to confirm or discard their new hypothesis (*Collins and Frank, 2013*; *Markant and Gureckis, 2014*). Hypothesis testing is known to be particularly valuable in environments with many choice options (*Gureckis and Markant, 2012*), but the fact that we observe it in a condition with only two options suggests that it is a constitutive feature of information seeking.

The notion of controllability bears a partial resemblance with the distinction between learning under selection vs. reception (*Gureckis and Markant, 2012*), and between free vs. forced choices (*Chambon et al., 2020*; *Ligneul, 2021*; *Sidarus et al., 2019*; *Wilson et al., 2014*). Performance benefits have been identified in selection contexts and free choice contexts, where participants can maximize the informativeness of their choices (*Freeman et al., 2014*; *Voss et al., 2010*; *Xu and Tenenbaum, 2007*). However, in these previous studies, sequences from selection and reception (and, likewise, from free and forced) choices vastly differ, which make it difficult to separate the specific effects of controllability from its consequences on the evidence available for subsequent choices. Instead, the current paradigm carefully matched the amount of evidence provided in both controllability conditions. Furthermore, we sought to validate controllability as the true cause of differences

between conditions. In Experiment 3, we examined whether a distinction in temporal focus (prospective instead of retrospective inference in an uncontrollable context) would account for the differences between the original conditions (*Figure 2—figure supplement 1*). Although the pattern of choices was markedly different, for confidence it remains possible that a lingering effect of temporality affected the original conditions, even if it cannot account for the results overall. In Experiment 2B, when the instructions changed unpredictably across trials instead of blocks, participants still perceived the controllable condition as more stable, providing evidence that the specificity of this condition arises from the controllability of information sampling experienced (*Figure 2—figure supplement 2*).

Despite stimuli carrying no explicitly affective or rewarding value, it remains possible that the mere presence of a target in the C+ condition makes the target category desirable and may produce a form of confirmation bias (*Talluri et al., 2018*). However, at the behavioral level, a confirmation bias would predict that in the controllable condition the sensitivity to evidence should be degraded when the sequence of evidence is inconsistent with participants' previous choice, a prediction that is absent from human data (*Figure 2*). At the neural level, a confirmation bias would decrease the weighting of belief-inconsistent evidence. Instead, in our previous study, we found an absence of difference between the neural coding of belief-consistent and belief-inconsistent evidence (*Weiss et al., 2021*). In addition, the findings of Experiment 2B, where the target category changes from trial to trial, also mean that the differences between conditions are unlikely to reflect a bias in direction of the target category (*Gesiarz et al., 2019*). Indeed, the direction of this bias would change from one trial to the next, and should therefore decrease in Experiment 2B – which we did not observe. Finally, based on theoretical considerations and empirical results, we also found evidence that our controllability manipulation did not create differences in working memory or executive demands between experimental conditions. We designed our conditions so that they were strictly matched in terms of number of tasks to do, sources of uncertainty, and motor actions to perform. At the behavioral level, the lack of a difference in choice accuracy, sensitivity to the objective evidence, and inference noise parameter between conditions makes it unlikely that the controllable condition was more demanding. At the physiological level, an increased load should have triggered changes in attention and arousal signals across all trials, unlike our observations that the neurophysiological measures only differed on switch decisions, whereas no difference was observed on repeat decisions. Together, these considerations make it unlikely that observed differences between conditions are due to an increased executive or working memory load in the controllable condition. At the neurophysiological level, we observed a stronger suppression of neuromagnetic alpha-band activity in the dorsal attention networks during the last seconds preceding and following response switches, in both conditions (*Figure 6*). This observation suggests that uncertainty at the time of a change-of-mind is associated with participants being more strongly oriented toward the presented evidence before the response switch occurs. Critically, this finding does not merely reflect greater attention or engagement in the controllable condition, which would have suggested general differences in task processing between conditions across repeat and switch decisions. Previous work using fMRI showed that task switching recruits a similar dorsal frontoparietal network involved in attentional control (*Corbetta and Shulman, 2002*), whereas response switching recruits a frontal network involved in executive control (*Daw et al., 2006*; *Donoso et al., 2014*; *Findling et al., 2019*). However, the use of MEG here allowed us to reveal largely anticipatory activations preceding response switches. Our results are also generally consistent with medial frontal activations during hypothesis testing (for a review, see *Monosov and Rushworth, 2022*) and in situations where the gathered information affords to predict but does not influence future outcomes (*White et al., 2019*). The temporal dissociation between switch effects observed in the occipital and frontal clusters suggests that changes-of-mind are preceded by a state of increased attention toward external (here, visual) evidence. This first stage is followed by a covert change-of-mind and ends with an overt response switch reflected in strong alpha-band suppression overlying frontal cortex (*Figure 6*, *Figure 6—figure supplement 1*).

In contrast to MEG dynamics, phasic pupil dilation revealed noradrenergic responses starting after the response probe onset and remained during presentation of the outcomes (*Figure 7*). This pattern of findings suggests a functional dissociation between alpha-band correlates of attention and pupil-linked arousal during changes-of-mind. The pupil-linked arousal associated with response switches is consistent with previous evidence for an association between pupil response and belief updating in uncertain environments (*Filipowicz et al., 2020*). Our results are also consistent with a previous study

providing evidence that pupil-linked arousal following decision uncertainty increased participants' tendency to switch their choice on the next trial (*Braun et al., 2017*). The fact that the pupil effect extends well into the next trial suggests that participants pay attention to the consequences of their changes-of-mind well into the next trial, particularly so in the controllable condition where they are in control of the upcoming sequence of evidence. This again stands in contrast to alpha suppression that was mostly anticipatory, a distinction between pupil dilation and alpha-band suppression previously observed in relation to attention (*Whitmarsh et al., 2021*). Even when accounting for the slower dynamics of pupillary responses (*Hoeks and Levelt, 1993*), the 4 s difference between the onsets of the effects observed in pupil dilation and alpha-band suppression makes it highly unlikely that the two effects arise from the same source. An additional difference between switch and repeat trials was further observed in the cardiac IBI, with similar dynamics as the alpha-band suppression. This increased slowing of heartbeats preceding response switches could partly reflect, but are unlikely to entirely reduce to, changes in respiration (*Park et al., 2014*), for example, participants holding their breath before response switches in the controllable condition. Together, these findings indicate that information seeking is characterized by a specific temporal succession of neurophysiological mechanisms in cortical, subcortical, and peripheral nervous systems.

By comparing changes-of-mind occurring in controllable and uncontrollable environments, we identified cognitive and neurophysiological signatures of information seeking that had otherwise proved difficult to isolate. We found that in controllable environments humans require more evidence to change their mind and do so with reduced confidence but are nevertheless more likely to stick to their decision on the next trial, suggesting a form of hypothesis testing. With computational modeling indicating that participants perceive controllable environments as more stable, we were not only able to explain why information seeking is associated with a higher degree of perceived uncertainty, but also to identify stronger and longer-lasting effects of changes-of-mind on cognition and behavior. Alterations in confidence (*Rouault et al., 2018*), information seeking (*Hauser et al., 2017*), and perceived controllability (*Voss et al., 2017*) are associated with various psychiatric symptoms. For instance, participants with obsessive-compulsive disorder typically need to gather more evidence before committing to a decision, while participants with schizophrenia present an inflated sense of agency associated with an inability to accurately update these representations (*Metcalfe et al., 2014*). By clarifying the effects of controllability on inference and confidence, our study lays the groundwork for explaining how the interplay between perceived control, information seeking, and confidence may go awry in psychiatric conditions.

## Materials and methods

### Participants

Human participants were recruited in the participant pool from the French platform 'Relay for Information about Cognitive Sciences' and provided written informed consent. The study was approved by the Comité de Protection des Personnes Ile-de-France VI, ID RCB: 2007-A01125-48, 2017-A01778-45. *Experiment 1*: 17 participants were originally recruited in February–March 2016. One participant was excluded for aborting the experiment before the end and one for performing at chance level, leaving 15 participants for behavioral analyses. *Experiment 2*: 20 participants were initially recruited in November–December 2016 and in March–May 2017. Two participants were excluded for performing at chance level, leaving 18 participants for behavioral analyses. *Experiment 3:* 30 participants were initially recruited in March 2019. Four participants were excluded for performing at chance level and one participant aborted the experiment before the end, leaving 25 participants for behavioral analyses. *Experiment 4*: 24 participants were tested in September–November 2015, whose behavior is described in *Weiss et al., 2021*.

### Behavioral tasks

#### Experiment 1

Participants performed a reversal learning task similar to that of our previous study (*Weiss et al., 2021*). Participants were presented with sequences of stimuli drawn from two discrete color categories (*Figure 1A*) and were asked to make a decision about the generative category on each trial (each sequence; *Figure 1B*). To examine a role for subjective confidence in relation to inference and

changes-of-mind about category, in addition to their choice, participants indicated their confidence (high or low) in their response using four response keys (**Figure 1B**).

Participants performed two experimental conditions that aimed at examining the influence of the degree of control over stimuli on choice and confidence. In both conditions, participants were required to track a hidden state (category). In the uncontrollable (C-) condition, participants were instructed that the computer draws sequences of stimuli and were asked to identify the category from which the stimuli were drawn (**Figure 1C**). An instruction screen indicated the mapping between response keys and color categories (counterbalanced across blocks). In the controllable (C+) condition, participants were instructed to draw stimuli from a given category (**Figure 1C**). An instruction screen indicated the target color category for each block (counterbalanced across blocks). In the C+ condition, participants have control over which stimuli to sample, but are not fully freely sampling, since they asked to produce stimuli from a target color category. Consequently, the hidden state differed between conditions: participants monitored changes in the category being drawn in the uncontrollable condition, whereas they monitored changes in the response key drawing from the target category in the controllable condition (**Figure 1D**). In both conditions, participants perform one action per sequence (after seeing a sequence in the uncontrollable condition, before seeing a sequence in the controllable condition). In other words, the uncontrollable condition requires monitoring the drawn category that flips occasionally, but does not require monitoring the category-action rule that is known and fixed over the course of the block. By contrast, the controllable condition requires monitoring the category-action rule that flips occasionally, but does not require monitoring the target category that is known and fixed over the course of the block. The conditions were therefore otherwise fully symmetric, tightly matched in terms of visual and motor requirements, and working memory demands. The order of condition administration was also counterbalanced pseudo-randomly across participants (C-/C+/C-/C+ for odd-numbered subjects, C+/C-/C+/C- for even-numbered subjects).

The generating category (hidden state) reversed occasionally and unpredictably at the same rate as in both conditions, with 'episodes' (i.e., chunks of trials) during which the hidden state was fixed (**Figure 1D**). Episode duration was sampled pseudo-randomly from a truncated exponential probability distribution (between 4 and 24 trials), resulting in a near-constant hazard rate in each block. Participants completed a total of 576 trials divided into blocks of 72 trials (72 sequences). The detailed instructions provided to participants are reported in **Weiss et al., 2021**. To experience the difference between experimental conditions, we provide a short gamified analog experiment at infdm.scicog.fr/aodemo/runner.html.

## Experiment 2

To examine whether the retention of a category goal over a longer time scale was critical in the participants' experience of control, and therefore influenced the results, we introduced a new rule manipulation. In half of the blocks, rules were stable across a block (hereafter, Experiment 2A). Since behavior was virtually identical in this rule condition, we then pooled these data with Experiment 1 for a total of 33 participants. In the other half of blocks, rules were changing on a trial basis instead of a block basis (hereafter, Experiment 2B). In the C- condition, participants were still asked to monitor the category from which samples were generated, but the action mapping (response keys) for reporting either category now changed on every trial. In the C+ condition, participants were still asked to generate samples from either category, but the target category now changed on every trial. The generative structure of the task and all other experimental features remained identical to Experiment 1. We analyzed this experimental data (2B) separately (**Figure 2—figure supplement 2**).

## Experiment 3

We reasoned that our C- and C+ conditions differ in terms of the degree of control that participants experience over stimuli. However, an alternative interpretation of these differences would be in terms of temporal orientation, with the C- condition being about monitoring past stimuli retrospectively (instruction to 'monitor'), whereas the C+ condition would be about producing stimuli prospectively (instruction to 'draw'). We set out to test this hypothesis using a modified version of the original conditions. Here, after each sequence of stimuli, participants were asked to guess from which category the computer drew from (retrospective condition) or to guess which category the computer will draw from (prospective condition). All other experimental features remained identical to Experiments 1 and 2A.

Findings related to choice behavior have been reported earlier (**Weiss et al., 2021**); we focus here on confidence analyses.

## Experiment 4

Detailed behavioral, modeling, and MEG analyses of this experimental data have been presented elsewhere (**Weiss et al., 2021**). In short, the experimental design was similar to that of Experiment 1 with the exception that no confidence responses were asked, only category choices.

## Stimuli

Stimuli were oriented bars presented on top of a colored circle displaying an angular gradient between orange and blue color categories spaced by π/2 (**Figure 1A**). Stimuli were drawn from either of these two categories (**Figure 1A**). On each trial, a sequence of 2, 4, 6, or 8 stimuli was drawn from a von Mises probability distribution centered on either category with a concentration of 0.5 and presented to participants for making a decision about category of origin at the end of the sequence (**Figure 1B**). Participants were not informed about sequence length on each trial, meaning that they had to pay attention to all stimuli within each sequence and could make up their mind until the time gap between the last stimulus of the sequence and the next probe. The number of stimuli per sequence was drawn from a uniform distribution and pseudo-randomized across trials. Stimuli were displayed at an average rate of 2 Hz with an inter-stimulus interval of 500 ± 50 ms. This rate is sufficiently low to perceive stimuli properly; here, the cognitive bottleneck was not on sensory processing, but on inference (**Drugowitsch et al., 2016**). The last stimulus of each sequence was followed by a longer delay of 1000 ± 50 ms, before participants were probed for their response.

## Statistical and quantitative analyses

### Psychometric analysis of behavior

We first examined two reversal curves: the proportion of choosing either option (**Figure 2A**) and the proportion of high-confidence responses (**Figure 2C**) as a function of trial number before and after a reversal. We modeled choices using a truncated exponential function characterized by two parameters, an asymptotic level and a (reversal) time constant (**Figure 2B**). For confidence as a function of reversal, we also fitted an exponential learning model for which we report two characteristic parameters (**Figure 2D**), a (confidence) time constant, and a 'confidence drop' parameter reflecting the drop between the lower ($p_{\min}$) and upper ($p_{\max}$) asymptotic levels (corrected by the time constant $t$), such that

$$\text{Confidence drop} = p_{\max} - p_{\min} + \left(p_{\max} - p_{\min}\right) \times \left(1 - e^{-1/t}\right)$$

We also examined the proportion of repeating the previous choice (**Figure 2E**) as a function of evidence recoded in favor of repeating a previous choice ('consistent evidence') or in favor of switching choice ('inconsistent evidence'). We quantified this by fitting a logistic function to quantify the amount of evidence required to switch a response in each condition (PSE) and the sensitivity to the evidence (slope of the sigmoid function; **Figure 2F**). Similarly, we analyzed the proportion of high-confidence responses as a function of consistent vs. inconsistent evidence (**Figure 2G**). For confidence, we computed within-participant error bars by removing the mean confidence level across conditions before computing the standard error. This was done to allow a fair comparison between conditions without an influence of inter-individual variability about the use and calibration of the high- and low-confidence responses. As expected, trials with consistent evidence correspond more often to repeat choices (right part of **Figure 2G**), whereas trials with inconsistent evidence correspond more often to changes-of-mind (left part of **Figure 2G**). To quantify these findings, we fitted two logistic sigmoid functions, one for repeat and one for switch choices, for each condition separately. The sensitivity to the evidence (slope) parameter was entered into a 2 × 2 ANOVA with factors CONDITION (C-, C+) and RESPONSE TYPE (repeat, switch). We also quantified the evidence level at which the two sigmoid for repeat and switch trials intersect, which corresponds to the quantity of evidence at which participants are equally confident in their repeat and switch decisions (**Figure 2H**). For all psychometric analyses, we compared parameters of best-fitting functions across conditions using paired $t$-tests. Wherever appropriate, we also performed Bayesian paired $t$-tests to assess the evidence in favor of the null

hypothesis using JASP version 0.8.1.2 with classic priors (zero-centered Cauchy distribution with a default scale of 0.707). When individual estimates were too noisy (e.g., for 6/33 participants, confidence was very little modulated by evidence level on switch trials), we performed a nonparametric Wilcoxon signed-rank test and displayed only the participants who did not have extreme values.

We additionally fitted the PSE on subsamples of trials corresponding to trials following a repeat and a switch decision, respectively (*Figure 5A*), and for trials following a high- or a low-confidence response, respectively (*Figure 5B*). This allowed us to examine how participants adapted their PSE on a dynamic basis, depending on the choice sequence experienced in the task.

## Change-of-mind analyses

In Experiments 1 and 2A, we sought to characterize the behavioral properties of changes-of-mind across conditions. In a 2 × 2 repeated-measures ANOVA, we examined the influence of RESPONSE TYPE (repeat, switch) and CONDITION (C-, C+) on the fraction of high-confidence responses (*Figure 3A*, which corresponds to the same data as *Figure 2G* pooled over objective evidence levels). For switch trials, we further examined confidence on switch trials that were confirmed on the next trial ('change-of-mind confirmed') compared to switch trials after which participants went back to their previous response ('change-of-mind aborted'; *Figure 3B*). Finally, we examined the fraction of changes-of-mind confirmed (over all changes-of-mind) as a function of whether the change-of-mind was done with high or low confidence (*Figure 3C*). These analyses are pooled over objective evidence levels due to each of these events not being distributed homogeneously across evidence levels.

To take into account any effects of evidence strength in these change-of-mind analyses, we further performed two logistic regressions (*Figure 3—figure supplement 1A and B*). First, we aimed to predict confidence using Response type, Condition, their interaction, and trial-by-trial evidence strength in favor of the previous response as a co-regressor (related to *Figure 3A*). Second, we aimed to predict confidence using change-of-mind (confirmed vs. aborted), Condition, their interaction, and trial-by-trial evidence strength in favor of the previous response (related to *Figure 3B*). We implemented regularized logistic regressions with Gaussian priors on each regression coefficient (mean = 0, SD = 2) in order to account for the fact that some of our participants have few events per cell, which otherwise led to unreliable regression coefficient estimates. Group-level significance of regression coefficients was assessed using one-sample *t*-tests. We also computed the average evidence quantity in favor of the participant's previous response and compared the obtained interactions patterns to those obtained in the change-of-mind analyses in *Figure 3* (*Figure 3—figure supplement 1C and D*).

## Computational model

### Model structure

We implemented a normative model of perceptual inference in volatile environments with contingency reversals (*Glaze et al., 2015*). A key aspect of the model is that it operates on the same evidence quantities and in a similar way in both C- and C+ conditions. Since previous work indicates that inference noise and not selection noise explains most of choice variability in such an inference task (*Drugowitsch et al., 2016*), we included no selection noise but we introduced inference noise on each sample of the sequence, which scales with sequence length (*Weiss et al., 2021*). We extended the model in key ways to predict not only choices, but also confidence. First, we introduced a confidence threshold parameter for determining whether the response will be provided with high or low confidence based on the posterior belief about the chosen category. This parameter captured baseline differences in proportion of high-confidence responses between conditions, representing the most parsimonious extension possible to provide a normative confidence response. Second, we introduced metacognitive noise to model an imperfect readout, as previously proposed (*Maniscalco and Lau, 2012*; *Pouget et al., 2016*). Third, we introduced a confidence gain parameter on switch trials modeling a differential readout of the posterior belief on changes-of-mind only as a multiplicative factor on the posterior belief applied on switch trials only.

### Model fitting

We fitted the model using a Bayesian Adaptive Direct Search (BADS) with 100 validation samples that provides point estimates for each parameter (*Acerbi and Ma, 2017*). We used five random starting points and parameters were bounded as follows (hazard rate: range = 0.000001–0.999999, inference

noise: range = 0–10, metacognitive noise: range = 0–10, confidence threshold: range=-10, +10, confidence gain for switches: range = 0–10). Here, we selected the most complete (parameterized) model, even at the risk of overfitting, because our goal was not to arbitrate between different models, but to compare our two conditions and our parameters under the same model. In addition, the behavioral effects associated with each of our parameters at the group level indicate that overfitting is unlikely to have occurred (see 'Model validation'). Indeed, if overfitting had occurred, meaning that if one of the parameters was essentially capturing noise, we would not observe consistent effects of a given parameter across participants (*Figure 4—figure supplements 3 and 4*).

We maximized the likelihood that model choices and confidence reproduce the reversal (*Figure 2*) and repetition (*Figure 3*) curves of participants, which are the key dimensions of interest for understanding participants' choice and confidence patterns. One participant was excluded for having an unreliable fit. All trials were fitted together, but each parameter was allowed to vary between conditions, and we compared the best-fitting parameters using paired *t*-tests at the group level. To ensure that our fitting procedure was unbiased, we performed a parameter recovery analysis (*Figure 4—figure supplement 1*). We simulated choice and confidence sequences using generative parameters randomly sampled in a uniform distribution between the minimum and maximum of participants' best-fitting parameter values in each condition. This procedure ensures that generative parameters were independently sampled. We then fitted these data using the same fitting procedure as for participants (except with three instead of five random starting points) and calculated the correlations between generative and recovered parameters, presented in a confusion matrix (*Figure 4—figure supplement 1*).

## Model validation

To validate the model, we simulated model choice and confidence from the best-fitting parameters on the same stimuli sequences as participants and analyzed the simulations similarly as for participants' data (*Figure 2—figure supplement 3*; *Palminteri et al., 2017*; *Wilson and Collins, 2019*). We examined correlations of the best-fitting parameters averaged across conditions between participants (*Figure 4—figure supplement 2*). We also performed a median split across participants on the best-fitting parameter values (*Figure 4—figure supplements 3 and 4*). We then averaged simulations of the model for each subgroup to further illustrate the independent contribution of each of these parameters. Note that we do not draw group inferences from these median split analyses – the goal is to visualize the effect of each of the parameters qualitatively.

Finally, to compare the characteristics of changes-of-mind between model and participants, we reproduced one of the psychometric analyses that fits the choice-PSE separately for trials following a repeat vs. a change-of-mind, and separately for trials following a high- vs. a low-confidence response. For model simulations, we averaged across 50 simulations per participant session (*Figure 5*).

## Magnetoencephalography

As reported in our previous study (Experiment 4, *Weiss et al., 2021*), we recorded MEG data using a whole-head Elekta Neuromag TRIUX system (Elekta Instrument AB, Stockholm, Sweden) composed of 204 planar gradiometers and 102 magnetometers using a sampling frequency of 1000 Hz. Prior to the experiment, each participant's head shape was digitized in the MEG coordinate frame. We used four additional head position indicator (HPI) coils, whose positions were also digitized, to monitor and correct for small head movements across blocks. The movement of HPI coils between blocks remained small (mean: 2.3 mm) was similar across blocks and across conditions ($t_{23}$ = 1.6, p=0.128).

We first removed magnetic noise from external sources using temporal signal space separation, after manually removing detected nonphysiological jumps in MEG signals. Stereotyped ocular and cardiac artifacts were corrected using a supervised principal component analysis (PCA). First, the onset of artifacts (either eye blinks or cardiac R-peaks) was automatically detected on auxiliary electrodes (EOG and ECG) synchronized with MEG signals using a threshold-based approach. MEG signals were then epoched from 100ms before to 400 ms after artifact onsets, and a PCA was used to extract the spatial components of cardiac and ocular artifacts. Typically, one stereotyped PCA component was removed from continuous MEG signals for eye blinks and two components for heartbeats.

The spectral power of band-limited MEG oscillations between 8 and 16 Hz was estimated using the 'multitapering' time-frequency transform implemented in FieldTrip (*Oostenveld et al., 2011*) (Slepian

tapers, eight cycles and three tapers per window, corresponding to a frequency smoothing of 25%), in two distinct time windows time-locked either to trial onset (from to 0 to +4s from trial onset) or to probe onset (from –4 to +4s from probe onset). We computed the contrast between repeat and switch decisions across conditions at each MEG channel, and examined pairwise correlations of these contrasts across all MEG channels. This analysis identified two main clusters, thereafter labeled 'occipital' and 'frontal' clusters based on their spatial localization. In these clusters, we employed jackknifed statistics to estimate temporal windows of significant differences between switch and repeat decisions, and between conditions (*Kiesel et al., 2008*).

## Pupillometry

To analyze physiological correlates of uncertainty across conditions, we measured pupil dilation at a sampling rate of 1000 Hz in Experiments 1 and 2. We preprocessed data with resampling at 100 Hz and corrected for blinks in a window between –100 ms and +500 ms around the event of interest using a derivative padding window between –200 ms and +200 ms and an instantaneous derivative threshold of 15. We smoothed the data with a moving average window size of 50 ms, detrended slow fluctuations from the signal using a characteristic time constant of 30 s, and z-scored the data. We excluded from the analyses blocks in which participants had more than 50% of low-quality data due to blinks (N = 2 participants excluded from these analyses).

We focused our analyses on phasic pupil RT-locked at the trial onset and response probe onset. We used an exclusion threshold of 2 s (if the window of missing data remained too large). For statistical analyses, we used a first-level uncorrected threshold of 0.05 and further performed correction for multiple comparisons with nonparametric cluster-level statistics computed across adjacent time points (*Maris and Oostenveld, 2007*). $t$, F, and p-values reported in the main text correspond to corrected statistical values based on permutation tests with 1000 permutations (*Figure 7B*). We report post hoc paired $t$-tests for comparing pupil dilation between conditions on repeat and switch trials separately in the time window identified after correction.

## Electrocardiography

We extracted the time points of cardiac R-peaks from ECG signals recorded continuously during Experiment 4 using an automatic threshold-based approach (z-score = 3) on band-pass-filtered signals between 1 and 16 Hz. We constructed time courses of cardiac IBI from these estimated R-peaks time-locked either to trial onset (from to 0 to +4 s from trial onset) or to response probe onset (from –4 to +4 s from probe onset). We analyzed these time courses similarly to time courses of alpha power suppression.

## Neurophysiological activity controlled for fluctuations in prior belief and evidence direction

To examine whether the differences between conditions visible in neurophysiological signals remain after controlling for our model variables, we analyzed alpha-band suppression time-locked at the trial onset and response probe onset in occipital and frontal clusters. We regressed out the effect of prior belief and evidence direction. We extracted these two variables from our computational model using particle filtering to only extract trajectories consistent with each participant's choice sequence (*Figure 8*). We performed analogous analyses for phasic pupil RT-locked at the response probe onset, and for time courses of cardiac IBI time-locked to the response probe onset.

## Acknowledgements

MR is the beneficiary of a postdoctoral fellowship from the AXA Research Fund. MR work was also supported by La Fondation des Treilles and department-wide grant from the Agence Nationale de la Recherche (ANR-17-EURE-0017, FrontCog). This work has received support under the program «Investissements d'Avenir» launched by the French Government and implemented by ANR (ANR-10-IDEX-0001-02 PSL). AW was supported by the FIRE Doctoral School. VC was supported by the French National Research Agency (ANR-16-CE37-0012-01 and ANR-19-CE37-0014-01). JD was supported by the James S McDonnell Foundation (grant #220020462). This work was supported by a starting grant from the European Research Council (ERC-StG-759341) awarded to VW, a junior researcher grant from the French National Research Agency (ANR-14-CE13-0028-01) awarded to VW, a France-US

collaborative research grant from the French National Research Agency (ANR-17-NEUC-0001-02) and the National Institute of Mental Health (1R01MH115554-01) awarded to VW and JD.

## Additional information

### Competing interests
Valentin Wyart: Reviewing editor, eLife. The other authors declare that no competing interests exist.

### Funding

| Funder | Grant reference number | Author |
|---|---|---|
| AXA Research Fund | | Marion Rouault |
| Agence Nationale de la Recherche | ANR-17-EURE-0017 | Marion Rouault Aurélien Weiss Junseok K Lee Valerian Chambon Valentin Wyart |
| Agence Nationale de la Recherche | ANR-10-IDEX-0001-02 PSL | Marion Rouault Aurélien Weiss Junseok K Lee Valerian Chambon Valentin Wyart |
| Agence Nationale de la Recherche | ANR-16-CE37-0012-01 and ANR-19-CE37-0014-01 | Valerian Chambon |
| James S. McDonnell Foundation | grant #220020462 | Jan Drugowitsch |
| European Research Council | ERC-StG-759341 | Valentin Wyart |
| Agence Nationale de la Recherche | ANR-14-CE13-0028-01 | Valentin Wyart |
| Agence Nationale de la Recherche | ANR-17-NEUC-0001-02 | Jan Drugowitsch Valentin Wyart |
| National Institute of Mental Health | 1R01MH115554-01 | Jan Drugowitsch Valentin Wyart |

The funders had no role in study design, data collection and interpretation, or the decision to submit the work for publication.

### Author contributions
Marion Rouault, Formal analysis, Investigation, Writing – original draft, Writing – review and editing; Aurélien Weiss, Data curation, Formal analysis, Investigation, Methodology; Junseok K Lee, Data curation, Investigation; Jan Drugowitsch, Investigation, Methodology, Writing – review and editing; Valerian Chambon, Data curation, Methodology, Writing – review and editing; Valentin Wyart, Conceptualization, Formal analysis, Supervision, Funding acquisition, Investigation, Methodology, Writing – original draft, Writing – review and editing

### Author ORCIDs

Marion Rouault http://orcid.org/0000-0001-6586-3788
Junseok K Lee http://orcid.org/0000-0002-8622-6259
Jan Drugowitsch http://orcid.org/0000-0002-7846-0408
Valentin Wyart http://orcid.org/0000-0001-6522-7837

### Ethics
Human subjects: Human participants were recruited in the participant pool from the French platform 'Relay for Information about Cognitive Sciences' and provided written informed consent. The study was approved by the Comité; de Protection des Personnes Ile-de-France VI, ID RCB: 2007-A01125-48, 2017-A01778-45.

Decision letter and Author response
Decision letter https://doi.org/10.7554/eLife.75038.sa1
Author response https://doi.org/10.7554/eLife.75038.sa2

## Additional files

### Supplementary files
• Transparent reporting form

### Data availability
Data and code availability. Participants' data and MATLAB code for behavioral and psychometric analyses of Experiments 1 and 2A, and code for fitting the computational model are available at https://www.github.com/marionrouault/actobscom/, (copy archived at swh:1:rev:43bfb417394bd243aa4c-cf2ee2b2f50850d4cea8). Because tested participants did not provide written consent regarding the posting of their anonymized data on public repositories, the MEG, pupillometric and ECG datasets are available from the corresponding authors upon request.

The following previously published dataset was used:

| Author(s) | Year | Dataset title | Dataset URL | Database and Identifier |
|---|---|---|---|---|
| Wyart V, Weiss A, Chambon V, Lee JK, Drugowitsch J | 2020 | Weiss_2020_NatComm_data_behavior | https://doi.org/10.6084/m9.figshare.13200128.v1 | figshare, 10.6084/m9.figshare.13200128.v1 |

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
