## [Editor Report]

This article will be of interest to psychologists and cognitive neuroscientists studying learning, decision-making, belief formation, and metacognition. The authors use an elegant task in which people make decisions with or without control over the information they sample, and link the cognitive processes at play to magnetoencephalography and pupillometry signatures. The key finding is that when participants have control over information sampling (i.e., they are seeking information), they need more contradictory evidence in order to switch their choices, and such switches are made with lower confidence, which is a clear conceptual advance in this field.

---

## [Decision Letter]

**Decision letter after peer review:**

Thank you for submitting your article "Controllability reveals defining features of information seeking" for consideration by *eLife*. Your article has been reviewed by 3 peer reviewers, and the evaluation has been overseen by a Reviewing Editor and Michael Frank as the Senior Editor. The following individual involved in review of your submission has agreed to reveal their identity: Clare Press (Reviewer #2).

The reviewers have discussed their reviews with one another, and the Reviewing Editor has drafted this to help you prepare a revised submission. The reviewers felt that the conclusions of of paper are mostly well supported by data, but some aspects of the effects reported, the model and psychological interpretation need to be clarified and extended.

Essential revisions:

The reviewers enjoyed this paper, which they found to be well written and with excellent figures, but felt that the number of different effects, psychological constructs, and data types presented at times impeded their understanding of a clear take-home message that captures how this paper furthers our understanding of decision-making. Any changes that could help focus the manuscript better in this regard would be welcome. Additionally, there are several re-analyses and changes to the interpretations that are required as detailed below, as well as some further details relating to the analyses. In principle the reviewers think that it should be possible to address these concerns without further data, but if it is more straightforward to conduct further experiments to bolster their conclusions then the authors should feel free to do that.

Key re-analyses required

1) The Ob trials require selection of an action, monitoring the action-outcome relationship, and a judgement about the stimulus. The Cb trials, in contrast, simply require monitoring the stimulus (it was unclear whether participants perform an action to start each trial, but regardless, they will not need to keep track of action-outcome mappings). Therefore, the fact that participants are slower to notice the shift in Ob trials and that their judgements are associated with lower confidence could easily be driven by the fact that they have three tasks (or at least two) rather than one. Relatedly, can the fact that α suppression and pupil dilation effects are increased with changes-of-mind in the Ob condition also be explained by the fact participants have more tasks? If there are greater executive/working memory demands, this will reduce α and increase dilation (as already discussed in the manuscript). Participants may detect stimulus changes less readily because there is a greater executive/working memory load in this condition, and this has nothing to do with controllability/action or stability of beliefs per se.

The authors claim that the participants were "equally sensitive" to the available objective evidence across conditions, which would indeed be reassuring, but at the moment it is not clear how solid this inference is. It would be good for the authors to clarify what they mean here – did they test for a significant slope/precision effect in the data shown in Figure 3A? Is it possible to say participants are equally sensitive to the evidence (other than on switch trials), when the insight into their sensitivity is given by a response that only requires them to process the evidence on switch trials? More generally, it remains to be convincingly demonstrated that the Ob-Cb differences are generated by something other than number of tasks and generic working memory/executive differences. For example, it would be important to demonstrate that performance accuracy (i.e. how often participants make the correct choice), and choice reaction times are equivalent between conditions, which could be achieved using a Bayesian approach. (This analysis would need to exclude the switch trials). If there are differences in performance accuracy or reaction times across conditions, then this could undermine the authors' conclusions.

2) In experiment 3, which aimed to test whether the temporal direction of the inference (prospective vs retrospective) could in fact explain the observed differences between condition, the effects on change of mind are not shown. Instead, only the effects on confidence are displayed – why is that? Additionally, the confidence data presented in Figure S1 still shows some apparent differences: lower overall confidence, higher confidence time-constant, and higher confidence-PSE in the prospective vs retrospective condition. Those differences are reported but the authors then nonetheless conclude that these results constitute evidence that the observed differences between Ob and Cb conditions in the main experiment are only due to controllability and not to the temporal orientation. At present, this conclusion is not supported by the data. One could conclude that temporality has no effect only if no difference in choice or confidence were observed. Again, a Bayesian approach to demonstrate equivalence could be of use here.

Introduction

3) Please unify, or at least operationalize more clearly, the terms that are used to describe the main effect of interest. Is information seeking (which indeed is often studied in bandit-type tasks) the same as changes of mind (which are more often studied in within-trial evidence accumulation paradigms)? How does this relate to belief updating, hypothesis testing, exploration, and the observers' stickiness tendency? While it is laudable that the authors reach across subfields of neuroscience that have their own terminology, in places the text was confusing about what's a general psychological process vs. an effect in the behavioral data vs. a model-derived computational latent variable.

4) Related to the above point, some of the terminology or logical formulation of conclusions was a bit confusing. The authors use the term "information-seeking" specifically to refer to the controllable condition. In the uncontrollable condition, there is no "seeking" of information, only passive sampling. If that understanding is correct, some sentences are redundant. For example, shouldn't the title read "Controllability reveals defining features of information-sampling" or even (and more consistent with the findings) "Controllability over information sampling reveals defining features of changes of mind"? At times "controllability" and "information-seeking" seem to be used interchangeably, while at other times they seem to have different meanings. It would be important to specify those terms better.

5) It would greatly help the reader to include a short paragraph in the introduction listing the ways in which this work replicates vs. extends the previous paper by Weiss et al.

6) Overall the introduction is a little thin and would benefit from being expanded. First, the statement that "information-seeking has been mostly studied under 'exploration-exploitation' dilemma is not true – there are many recent studies that have studied information-seeking in humans using other paradigms. Second, and most importantly, the introduction currently lacks the rationale for the proposed work – in addition to dissociating information-seeking from changes-of-mind, which is more of a methodological aim, what are the key questions that the authors trying to address, and what are the hypotheses given the current literature? Finally, the use of MEG and other physiological measures (pupillometric, cardiac patterns) is not motivated at all in the introduction. The introduction should set the stage as to why collecting these data is needed given the question of interest. Similarly, the hypotheses should also set the stage for the specific analysis choices performed later, i.e. in what way does each presented analysis answer the question?

Methods

7) Was the model fitted to all trials together, or separately for each condition?

8) The model validation is rigorous, but would the same patterns also be observed out-of-sample? Since only one model is tested, there is always a risk of overfitting, which could be addressed with out-of-sample validation.

9) Please provides the statistics for the validation data presented on p12 (Figure S2 and S3)

10) How recoverable are the parameters if the model is re-fit to simulated data?

11) Are parameters correlated with each other? The authors should provide a correlation table of the parameters in the supplement.

Results

12) It would help to preview the specific diagnostic effects that isolate the process of interest, and then to repeat those in the model fit. Currently, we see four effects in Figure 2, four effects in Figure 3, and three effects in Figure 4. When we get to Figure 5D, slightly different effects are presented in comparison to the model. This was quite confusing and more clarity is required about the specific behavioral pattern of interest. This may just be a matter of slight reordering: the supplement has clear model-data matches, and placing those in the main figure (to highlight that most of the basic effects in Figures2-4 are indeed explained by the model) would allow a better appreciation of where the model fails to capture human behavior.

13) (see also (2) above) The data in Figure S1 are a crucial control, but it is not yet clear that there is no substantial difference in pro- vs. retro-spective decisions, which might (partially) explain the Cb vs. Ob differences. If controllability was the only important factor for behavioral effects, wouldn't we expect no difference between conditions in Figure S1? This would require a change in the authors' strength of interpretation as specifically pertaining to control.

14) Do the analyses in Figure 4 control for the difference in objective evidence (logL per trial)?

15) It is frequently stated that the Ob condition bestows instrumental control to participants over the sampling of stimuli. These statements should be modified, given that participants are told which category of stimulus they must generate. They do not have complete control over which stimuli they sample, unless they are willing to upset the experimenter!

16) It is not clear what action participants perform to start the trial in the Ob condition. This needs including. In outlining this it would also be great to include consideration of potential overlaps between the responses at the start and end of the trial in Ob, and whether this might have played into the effects.

17) Please add the change-of-mind data (i.e. equivalent of Figure 2A-B and 3A-B) for experiment 3, so it is possible to dissociate which effects are unique to the main experiment and which effects may be due to the temporal direction of the decisions. It is important to clarify this distinction as it can actually help narrow down the unique effect of controllability over temporality.

18) It would be useful to provide follow-up analyses to determine how much of the behavioral results can be explained by the difference in prior belief reported on p19, as well as discuss how much of these findings are then more consistent with a role of controllability per se, rather than a role for motivated beliefs/confirmation bias due to the presence of a target.

Discussion

19) There is some concern about the interpretation of the results – in particular, whether the observed differences between the cue-based and outcome-based conditions could be better explained by the presence of a target-induced confirmation bias in the outcome-based condition, which would induce a strong motivation to confirm that the chosen action does lead to the target. In other words, it is possible that having a target (e.g. "draw orange") may bias subjects towards believing they are drawing from that target even when they are not, and as a result needing less evidence to reach that conclusion (similar to the effect shown in Gesiarz, Cahill and Sharot, 2019, Plos Computational Biology). This could in turn lead to the observed patterns of results, i.e. needing more evidence against orange to switch, being less confident when switching etc. Additionally, the result that prior beliefs are stronger in the Ob condition (p18-19) is consistent with this idea, since confirmation bias is usually associated with stronger prior beliefs.

20) The authors claim that there is a causal role for confidence in controlling changes-of-mind (p11). While this is an interesting idea, it is not clear that it can be fully evidenced in this task without an experimentally-controlled manipulation of confidence. The reason is that there could be a common cause to both high confidence and high propensity to confirm switch decisions without the two processes actually being causally related. One such common cause could be the strength of evidence. Therefore, this conclusion needs to be tempered and this issue mentioned in the discussion.

21) There is no discussion of alternative ways in which these data might have turned out. The discussion entirely reports claims and findings with which the present data are consistent. Are there no data or theories that could have led to alternative predictions? Currently the manuscript portrays the impression of such high consensus that there was little point running the studies, suggesting that the empirical patterns to date all point in identical directions. One possibility is that the stimulus ISIs are not very variable. Given participants can perceive better at certain oscillatory phases of visual processing (peaks), could they choose to start trials in Ob to align with peaks and thereby improve perceptual acuity? If this occurred, the influence must be weaker than those pulling in the opposite direction, but it could have led to an alternative outcome. If it's a straightforward analysis to perform/report, it may be interesting to see how the oscillations align with events differentially in Ob and Cb.

---

## [Author Response]

Essential revisions:The reviewers enjoyed this paper, which they found to be well written and with excellent figures, but felt that the number of different effects, psychological constructs, and data types presented at times impeded their understanding of a clear take-home message that captures how this paper furthers our understanding of decision-making. Any changes that could help focus the manuscript better in this regard would be welcome. Additionally, there are several re-analyses and changes to the interpretations that are required as detailed below, as well as some further details relating to the analyses. In principle the reviewers think that it should be possible to address these concerns without further data, but if it is more straightforward to conduct further experiments to bolster their conclusions then the authors should feel free to do that.

We thank both the editor and reviewers for their positive assessment of our paper, and for their helpful and thorough comments, which had a substantial impact on the manuscript. Following the editor and reviewers’ feedback:

We have re-written our introduction to clarify: 1. how the paper contributes to the characterisation of information seeking in decision-making under uncertainty by positioning it with respect to the previous literature on this topic, and 2. the relationship between information seeking and the two related concepts of exploration and changes-of-mind;We now present more data from Experiment 3 to address the possibility that our main findings could be interpreted in terms of temporal orientation (retrospective vs. prospective inference) rather than controllability;We use several lines of evidence to show that the effects observed in the controllable condition that we attribute to information seeking cannot be explained by higher cognitive demands and/or working memory load;Please note that upon the reviewers’ suggestion, we have now renamed the conditions: uncontrollable condition (C-) (ex. cue-based condition) and controllable condition (C+) (ex. outcome-based condition).

Below we describe how we have revised our manuscript in response to each individual comment.

Key re-analyses required1) The Ob trials require selection of an action, monitoring the action-outcome relationship, and a judgement about the stimulus. The Cb trials, in contrast, simply require monitoring the stimulus (it was unclear whether participants perform an action to start each trial, but regardless, they will not need to keep track of action-outcome mappings). Therefore, the fact that participants are slower to notice the shift in Ob trials and that their judgements are associated with lower confidence could easily be driven by the fact that they have three tasks (or at least two) rather than one. Relatedly, can the fact that α suppression and pupil dilation effects are increased with changes-of-mind in the Ob condition also be explained by the fact participants have more tasks? If there are greater executive/working memory demands, this will reduce α and increase dilation (as already discussed in the manuscript). Participants may detect stimulus changes less readily because there is a greater executive/working memory load in this condition, and this has nothing to do with controllability/action or stability of beliefs per se.

We understand the reviewer’s request for clarifications regarding the paradigm. From a computational standpoint, and although the uncontrollable (C-) and controllable (C+) conditions are different in that they require participants to track different hidden states, they do not differ in the number of cognitive operations (or ‘tasks’) that need to be performed to solve them. We now use model simulations to show that the observed differences between the two conditions are inconsistent with the idea that participants have more cognitive operations to perform in the C+ condition. Finally, in response to the second part of the reviewer’s comment below, we provide new analyses of the behavioral data that speak against a higher load in the C+ condition. We hope that these different additions have clarified why the observed differences between the two conditions cannot be attributed to a difference in executive or working memory load.

In both conditions, participants track a single hidden state, but the nature of this hidden state differs between conditions. Each condition requires tracking a single hidden state of the task: the category (A or B) drawn by the computer in the uncontrollable condition, and the action (left or right) drawing the target category in the C+ condition. In both conditions, participants need to select an action for each stimulus sequence. In the uncontrollable condition, participants select the action most likely to be associated with the category being drawn by the computer, based on a category-action rule defined at the beginning of the current block (A-left and B-right or vice versa). In the C+ condition, participants select the action most likely to be associated with the target category defined at the beginning of the current block. In other words, the uncontrollable condition requires monitoring the drawn category which flips occasionally, but does not require monitoring the category-action rule which is known and fixed over the course of the block. By contrast, the C+ condition requires monitoring the category-action rule which flips occasionally, but does not require monitoring the target category which is known and fixed over the course of the block. Importantly, in both conditions, each trial (sequence of stimuli) is associated with a single action (key press). Therefore, and while the two conditions are indisputably different, because participants have instrumental control over the category being drawn in the C+ condition, they do not differ in the number of task variables that need to be monitored.

We have now rewritten the protocol in our Methods section to make this point clearer (p. 27-28):

“Participants performed two experimental conditions that aimed at examining the influence of the degree of control over stimuli on choice and confidence. In both conditions, participants were required to track a hidden state (category). In the uncontrollable (C-) condition, participants were instructed that the computer draws sequences of stimuli, and were asked to identify the category from which the stimuli were drawn (Figure 1C). An instruction screen indicated the mapping between response keys and color categories (counterbalanced across blocks). In the controllable (C+) condition, participants were instructed to draw stimuli from a given category (Figure 1C). An instruction screen indicated the target color category for each block (counterbalanced across blocks). Consequently, the hidden state differed between conditions: participants monitored changes in the category being drawn in the uncontrollable condition, whereas they monitored changes in the response key drawing from the target category in the controllable condition (Figure 1D). In both conditions, participants perform one action per sequence (after seeing a sequence in the uncontrollable condition, before seeing a sequence in the controllable condition). In other words, the uncontrollable condition requires monitoring the drawn category which flips occasionally, but does not require monitoring the category-action rule which is known and fixed over the course of the block. By contrast, the controllable condition requires monitoring the category-action rule which flips occasionally, but does not require monitoring the target category which is known and fixed over the course of the block. The conditions were therefore otherwise fully symmetric, tightly matched in terms of visual and motor requirements, and working memory demands. The order of condition administration was also counterbalanced pseudo-randomly across participants (C-/C+/C/C+ for odd-numbered subjects, C+/C-/C+/C- for even-numbered subjects).”

Beyond the structure of the paradigm, there are also several aspects of the behavioral and neurophysiological data that are inconsistent with the idea of an increased executive or working memory load in the C+ condition. We agree with the reviewers that reduced attention and/or increased load due to the larger number of tasks to perform in the C+ condition would translate into slower behavioral adaptation to hidden-state reversals, as observed in the C+ condition. However, this alternative account of the behavioral effect would also produce a *decrease* in the asymptotic reversal rate in the C+ condition – a prediction which conflicts with the *increased* asymptotic reversal rate observed in this condition (Figure 2B).

To support this argument, we ran model simulations implementing the idea of an increased working memory load. Specifically, we reasoned that increased working memory load would result in attentional lapses, meaning that either a fraction of stimuli within a sequence would be missed (variant 1), or a fraction of sequences would be missed (variant 2). We implemented such a model with attentional lapses corrupting inference at the level of stimuli (variant 1) or sequences (variant 2; Author response image 1). We started from best-fitting parameters in the C- condition, and added attentional lapses to simulate behavior in the C+ condition.

**Author response image 1. sa2fig1:** Model simulations of the original uncontrollable condition (blue) and an uncontrollable condition further altered by an increased working memory demand (purple). Attentional lapses were implemented at the level of the stimulus within a sequence (variant 1, top row) or at the level of the whole sequence (variant 2, bottom row). Both forms of attentional lapses resulted in a slower adaptation after reversals (left column) and a decreased sensitivity to evidence (right column), all inconsistent with participants’ choice patterns observed in the C+ condition.

At the level of stimuli (variant 1, top row), model simulations indicate that 60% of stimuli would need to be missed (i.e., ignored) to obtain a choice-PSE – i.e., the quantity of conflicting evidence needed to switch – similar to the one we observed in the C+ condition. This fraction of attentional lapses seems highly unlikely.

As expected, adaptation to reversals is slower in the presence of lapses. Psychometric fits of model simulations are qualitatively inconsistent with those of human participants: with lapses, the asymptotic reversal rate is strongly decreased relative to the C- condition (without lapses). At the level of sequences (variant 2, bottom row), a similar pattern is obtained: 30% of sequences would need to be missed to obtain a choice-PSE similar to that of the one observed in the C+ condition, which again seems highly unlikely. As for stimulus-level lapses (variant 1), sequence-level lapses produce a slower adaptation to reversals and a reduced asymptotic reversal rate, an effect inconsistent with participants’ behavior in the C+ condition (Figure R1). Together, these patterns are therefore inconsistent with an interpretation that the C+ condition required an increased demand.

Furthermore, attentional lapses due to increased working memory load would decrease the sensitivity to the available objective evidence – a prediction which conflicts with the similar sensitivity found in the two conditions (Figure 2F). An increased working memory load would also predict slower response times (RTs) in the C+ condition on repeat decisions, which we did not observe in our data (please see response to the next comment for a full breakdown). Together, these different considerations make it highly unlikely that observed differences between conditions are due to an increased executive or working memory load in the C+ condition.

Last, there are important aspects in physiological data that are inconsistent with an increased load in the C+ condition. Indeed, an increase in working memory load (due to more tasks to perform in the C+ condition) should trigger increased arousal and attentional signals across all trials in the C+ condition. This prediction does not match our observations (Figure 6F and 7B): indeed, α-band power and pupil dilation do not differ between conditions for trials ending with a repetition of the previous response (the majority of the trials). These two neurophysiological measures differ between conditions only for trials ending with a switch from the previous response (i.e., the minority of trials associated with information seeking). These selective differences between conditions for switch trials only are again inconsistent with the idea of an overall increase in executive or working memory load in the C+ condition.

We have now added a new paragraph in the Discussion section to describe – and rule out – this alternative account of the differences between conditions (p. 24):

“Finally, based on theoretical considerations and empirical results, we also found evidence that our controllability manipulation did not create differences in working memory or executive demands between experimental conditions. We designed our conditions so that they were strictly matched in terms of number of tasks to do, quantities to monitor, sources of uncertainty, and motor actions to perform. At the behavioral level, the lack of a difference in choice accuracy, sensitivity to the objective evidence, and inference noise parameter between conditions makes it unlikely that the C+ condition was more demanding. At the physiological level, an increased load should have triggered changes in attention and arousal signals across all trials, unlike our observations that the neurophysiological measures only differed on switch decisions, whereas no difference was observed on repeat decisions. Together, these considerations make it highly unlikely that observed differences between conditions are due to an increased executive or working memory load in the C+ condition.”

The authors claim that the participants were "equally sensitive" to the available objective evidence across conditions, which would indeed be reassuring, but at the moment it is not clear how solid this inference is. It would be good for the authors to clarify what they mean here – did they test for a significant slope/precision effect in the data shown in Figure 3A? Is it possible to say participants are equally sensitive to the evidence (other than on switch trials), when the insight into their sensitivity is given by a response that only requires them to process the evidence on switch trials? More generally, it remains to be convincingly demonstrated that the Ob-Cb differences are generated by something other than number of tasks and generic working memory/executive differences. For example, it would be important to demonstrate that performance accuracy (i.e. how often participants make the correct choice), and choice reaction times are equivalent between conditions, which could be achieved using a Bayesian approach. (This analysis would need to exclude the switch trials). If there are differences in performance accuracy or reaction times across conditions, then this could undermine the authors' conclusions.

Upon the reviewer’s suggestions, we have performed several additional analyses that provide strong evidence against a difference in executive demands or working memory load between conditions, which we develop below.

*1) Sensitivity to the evidence.* Two findings support the claim that participants were equally sensitive to the available objective evidence across conditions. It is important to note that participants had to process the evidence provided by stimuli on *all* trials, not only on switch trials. First, as suggested by the reviewer, we have tested for a difference in slope in the data shown in Figure 2E/F. We found no difference in this slope parameter (*t*_32_=1.09, *p*=.28), as displayed on (ex-Figure 3B) as the ‘sensitivity to evidence’ parameter:

Second, in our computational model, the best-fitting inference noise parameter was not significantly larger in the C+ condition (*t*_32_=1.45, *p*=.16; Figure 4B). This parameter corresponds to a model-based equivalent of the variability of participants’ decisions with respect to the objective evidence presented (inverse of sensitivity to evidence). Together, these results indicate that objective evidence was integrated equally well across experimental conditions.

*2) Choice accuracy.* There was a difference in choice accuracy (i.e., how often participants make the correct choice) between conditions (*t_32_*=-3.103, *p*=.004). Importantly however, the difference was in the direction of a *worse* choice accuracy in the C- (78.74% correct) than in the C+ (80.57% correct) condition. This result therefore goes against the alternative interpretation raised by the reviewer that the C+ condition would create more tasks and more cognitive demands, hence a lower choice accuracy. Instead, the performance of participants in the C+ condition being not degraded and actually a bit higher suggests that participants do not miss significantly more information in this condition.

Despite being significant in Experiments 1 and 2A, the magnitude of the difference in choice accuracy is small (a 2% increase). Moreover, across datasets, this result was not consistent. Indeed, no difference in choice accuracy was found in Experiment 2B (*t_17_*=.58, *p*=.569). No difference in choice accuracy was found in Experiment 4 (*t_23_*=0.3, *p*=.784) (Weiss et al., 2021). In an ongoing online version of the task (unpublished) that corresponds to the experimental design of Experiments 1 and 2A, we found no difference in choice accuracy between conditions with *N*=200 participants (*t_199_*=-1.46, *p*=0.145; C-: mean=77.06% correct; C+: mean=77.96% correct). Together, these observations provide further evidence against significant differences in working memory demands between conditions.

*3) Response times.* Upon the reviewer’s suggestion, we compared RTs between conditions on repeat trials. We found similar RTs between C- and C+ condition (*t_32_*=-1.52, *p*=.14). These results indicate that the differences between conditions are unlikely to be generated by working memory or executive differences. Using JASP, we further performed a Bayesian t-test which provided evidence for a genuine lack of a difference between C- and C+ conditions (repeat decisions: BF_10_=0.531). Together, these findings provide additional evidence that the working memory demands were very similar across experimental conditions, and that participants did not perform more tasks in the C+ condition (Results section, p. 7):

“Participants’ choice accuracy was higher in the C+ (80.6% correct) than in the C- (78.7% correct) condition (t_32_=-3.103, p=.004), although this difference was only of 2% correct. Moreover, response times (RTs) on repeat decisions were similar between C- and C+ condition (t_32_=-1.52, p=.14, BF_10_=0.531). These initial results indicate that the differences between conditions are unlikely to be generated by working memory or executive demand differences.”

*4) Additional condition.* Experiment 2B includes a manipulation which effectively changes the number of tasks that need to be performed. We changed the category-action mapping randomly from one trial to the next in the C- condition, and changed the target category randomly from one trial to the next in the C+ condition. This variant requires participants to account for the rule provided on each trial, rather than relying on a fixed rule across trials in the original variant of the two conditions (Experiments 1 and 2A). To measure the effects of an increase in the number of tasks that need to be performed, we compared task metrics between Experiments 2A and 2B variants of the two conditions (done by the same participants). Importantly, we found that the choice-PSE did not increase in the conditions of Experiment 2B where more tasks need to be performed as compared to the conditions of Experiment 2A (C-: *t_17_*=.77, *p*=.45; C+: *t_17_*=.73, *p*=.47). Furthermore, the difference in choice-PSE between the C- and C+ conditions did not differ between Experiments 2A and 2B (*t_17_*=.49, *p=*.63). Together, these additional findings show that increasing the number of tasks that need to be performed does not alter the choice-PSE in either condition. Therefore, even if participants performed more tasks in the C+ condition (something which is highly unlikely based on the evidence detailed above), it is unlikely that this difference would cause the larger choice-PSE in the C+ condition, or the selective change in pupil dilation and α power only in trials ending with response switches (i.e., where participants engage in information seeking).

We now report these new findings in the Results section (p. 10):

“Importantly, even if Experiment 2B effectively changes the number of tasks that need to be performed, we observed a similar Choice-PSE between Experiments 2A and 2B (C-: t_17_=.77, p=.45; C+: t_17_=.73, p=.47), and the difference in choice-PSE between the C- and C+ conditions did not differ between Experiments 2A and

2B (t_17_=.49, p=.63).”

In our Discussion section, we now detail our new analyses and findings regarding the working memory demands of the two conditions (p. 24):

“Finally, based on theoretical considerations and empirical results, we also found evidence that our controllability manipulation did not create differences in working memory or executive demands between experimental conditions. We designed our conditions so that they were strictly matched in terms of number of tasks to do, quantities to monitor, sources of uncertainty, and motor actions to perform. At the behavioral level, the lack of a difference in choice accuracy, sensitivity to the objective evidence, and inference noise parameter between conditions makes it unlikely that the C+ condition was more demanding. At the physiological level, an increased load should have triggered changes in attention and arousal signals across all trials, unlike our observations that the neurophysiological measures only differed on switch decisions, whereas no difference was observed on repeat decisions. Together, these considerations make it highly unlikely that observed differences between conditions are due to an increased executive or working memory load in the C+ condition.”

2) In experiment 3, which aimed to test whether the temporal direction of the inference (prospective vs retrospective) could in fact explain the observed differences between condition, the effects on change of mind are not shown. Instead, only the effects on confidence are displayed – why is that? Additionally, the confidence data presented in Figure S1 still shows some apparent differences: lower overall confidence, higher confidence time-constant, and higher confidence-PSE in the prospective vs retrospective condition. Those differences are reported but the authors then nonetheless conclude that these results constitute evidence that the observed differences between Ob and Cb conditions in the main experiment are only due to controllability and not to the temporal orientation. At present, this conclusion is not supported by the data. One could conclude that temporality has no effect only if no difference in choice or confidence were observed. Again, a Bayesian approach to demonstrate equivalence could be of use here.

We thank the reviewer for raising the important point of the nature of the difference between the C- and C+ conditions, whether controllability or temporal orientation (prospective vs. retrospective). We now also present the choice data in addition to the confidence data from Experiment 3, which we adapted from Weiss et al., 2021 and included in a new Figure 2 supplement 1. Critically, the findings of Experiment 3 reveal that the nature and magnitude of the contrast between retrospective and prospective conditions is different from the contrast between the original C- and C+ conditions, and therefore cannot account for them as an overall explanation of the results, even if a remaining effect of temporal direction of the inference might still be present in the C- vs. C+ conditions, which we now acknowledge in our Discussion section. We also expand below on the differences and similarities between Experiment 3 and the main Experiments 1 and 2A.

*Choices*. In contrast to the comparison between C- and C+ conditions, there were no significant differences in choice adaptation after a reversal between retrospective and prospective conditions. Participants’ reversal time constants were similar across retrospective and prospective conditions (*t_24_*=0.1, *p*=.98). Critically, the amount of inconsistent evidence required to change their mind (choice-PSE) was the same between retrospective and prospective conditions (*t_24_*=1.7, *p*=.109), unlike between the original C- and C+ conditions.

*Confidence*. First, there was no difference in overall confidence between retrospective and prospective conditions (*t_24_*=-1.2, *p*=.25). Moreover, the differences between conditions are not the same, of different direction. Even if there was a higher confidence time constant (*t_24_*=-2.58, *p*=.016), the effect is small and noisy: indeed, it depends on the presence of a confidence drop to be measured reliably. In Experiment 3, however, the confidence drop was not different from zero in both retrospective (*t_24_*=1.6, *p*=.12) and prospective (*t_24_*=.42, *p*=.67) conditions. Therefore, the confidence time constant should not be interpreted, and we no longer present it on Figure 2 supplement 1, due to the unreliability of its estimation. Second, even if the confidence-PSE is increased in the prospective as compared to the retrospective condition, the confidence drop is no longer different between retrospective and prospective conditions. Third, the sensitivity of confidence to the evidence on switch decisions, that was about twice larger in C- as compared to C+, is now similar between prospective and retrospective conditions (*t_24_*=-0.94, *p*=.35). This result indicates that participants were less confident both in the face of evidence consistent and inconsistent with their previous choice, which was not the case for the original C- and C+ conditions.

Taken together, these results indicate that even if a small temporality effect possibly remained present in Experiments 1 and 2A, it cannot alone be an overall explanation of our results, and cannot explain the main differences observed between the C- and C+ conditions as a whole, which we therefore ascribe mostly to controllability. We now include a Discussion point for acknowledging this alternative interpretation in our Discussion section (p. 23):

“Furthermore, we sought to validate controllability as the true cause of differences between conditions. In Experiment 3, we examined whether a distinction in temporal focus (prospective instead of retrospective inference in an uncontrollable context) would account for the differences between the original conditions (Figure 2 supplement 1). Although the pattern of choices was markedly different, for confidence it remains possible that a lingering effect of temporality affected the original conditions, even if it cannot account for the results overall.”

Introduction3) Please unify, or at least operationalize more clearly, the terms that are used to describe the main effect of interest. Is information seeking (which indeed is often studied in bandit-type tasks) the same as changes of mind (which are more often studied in within-trial evidence accumulation paradigms)? How does this relate to belief updating, hypothesis testing, exploration, and the observers' stickiness tendency? While it is laudable that the authors reach across subfields of neuroscience that have their own terminology, in places the text was confusing about what's a general psychological process vs. an effect in the behavioral data vs. a model-derived computational latent variable.

We thank the reviewer for prompting clarity on the meaning and terminology for the different effects of interest. We agree with the reviewer that the term ‘exploration’ in the literature has a heterogeneous meaning, and has been used to refer to a number of different cognitive processes.

We have typically used ‘changes-of-mind’ to refer to the cognitive operation, while ‘response switch’ and ‘switch decision’ refer to the behaviour observable by the experimenter. We consider information seeking to be only possible in the controllable condition, whereas in the uncontrollable condition, only passive sampling occurs. It is the control over evidence conferred to participants in the controllable experimental condition that renders information seeking possible. We have revised the manuscript to use appropriate terms throughout, and we have now updated Figure 1 about the paradigm to explain the interactions between the different experimental and cognitive events. Hidden-state reversals are determined by the experimenter (“task event”). During an exploratory decision, several steps co-occur: a covert change-of-mind (“cognitive event”) about the expected reward of the current option; an overt response switch (“behavioral event”); and information seeking, i.e., an active search for information about the new option being considered, which is only possible in the C+ condition where participants are in control. (Results section, p. 6):

Upon the reviewer’s suggestion, in our introduction section, we now clarify the relationship between information seeking and other related concepts such as belief updating, hypothesis testing and exploration:

“In these paradigms, participants evolve in controllable environments and usually sample one among several options to maximize reward. Therefore, they have to either exploit a currently rewarding option, or sacrifice rewards to explore alternative options and seek information about their possible rewards (Rich and Gureckis, 2018; Wilson et al., 2014). This trade-off means that in these paradigms, exploration differs from exploitation not only in terms of information seeking, but also in terms of other co-occurring cognitive events, including overt response switches and covert changes-of-mind.

These different families of paradigms developed for studying information seeking vary on several dimensions, particularly the sources of uncertainty (Fleming et al., 2018), the stimuli used (Gesiarz et al., 2019), the desirability of the information to be sought (Hertwig et al., 2021), and the degree of control over information sampled (Desender et al., 2018). These differences have made direct comparisons between paradigms extremely challenging. To date, no study has directly manipulated control over evidence sampling in otherwise aligned experimental conditions.” (p. 3-4)

4) Related to the above point, some of the terminology or logical formulation of conclusions was a bit confusing. The authors use the term "information-seeking" specifically to refer to the controllable condition. In the uncontrollable condition, there is no "seeking" of information, only passive sampling. If that understanding is correct, some sentences are redundant. For example, shouldn't the title read "Controllability reveals defining features of information-sampling" or even (and more consistent with the findings) "Controllability over information sampling reveals defining features of changes of mind"? At times "controllability" and "information-seeking" seem to be used interchangeably, while at other times they seem to have different meanings. It would be important to specify those terms better.

We thank the reviewer for prompting clarification on the concepts manipulated. The reviewer is correct that we consider information seeking to be only possible in the controllable condition, whereas in the uncontrollable condition, only passive sampling occurs. We define controllability as the manipulation created by the experimenter. It is the control over evidence conferred to participants in the controllable experimental condition that renders information seeking possible. We have revised the manuscript so as to use the right terms for the right meaning throughout, and we have now updated Figure 1 about the paradigm to explain the interactions between the different experimental and cognitive events. Hidden-state reversals are determined by the experimenter (“task event”). During an exploratory decision, several steps co-occur: a covert change-of-mind (“cognitive event”) about the expected reward of the current option; an overt response switch (“behavioral event”); and information seeking, i.e., an active search for information about the new option being considered, which is only possible in the C+ condition where participants are in control.

Accordingly, following upon the reviewer’s suggestion, we have now changed the title of the study to “Controllability boosts neural and cognitive signatures of changes-of-mind in uncertain environments”, a title now focusing more specifically on our results.

Although we fully agree with the reviewer’s statement that control is necessary for information seeking, other authors have reasoned otherwise, based on other definitions of information seeking. In particular, Monosov and Rushworth distinguish the notions of ‘passive’ and ‘active’ information seeking (Bromberg-Martin and Monosov, 2020 *Current Opinion in Behavioral Science*; Monosov and Rushworth, 2022 *Neuropsychopharmacology*). The authors refer to decision-making tasks under uncertainty, in which non human primates typically can select to sample information about potential rewards, even if the information sampled typically has no direct instrumental value, and does not confer them control over these rewards (e.g., White et al., 2019 *Nat Commun*). Such ‘non instrumental information sampling’ or ‘passive information seeking’ is proposed to be valuable in and of itself if acquiring this information might be useful in the future for guiding behavior – i.e., that might become instrumental later on in the long run. However, our definition of information *seeking* is tightly coupled to the presence of active control, in line with what other authors have proposed (Gureckis and Markant, 2012 *Psychological Science*). We refer to the notion of non-instrumental information *sampling* (not *seeking*) in our Discussion section:

“Our results are also generally consistent with medial frontal activations during hypothesis testing (for a review, see Monosov and Rushworth, 2022) and in non-instrumental information sampling, where the gathered information affords to predict but does not influence future outcomes.” (Discussion section, p. 24)

5) It would greatly help the reader to include a short paragraph in the introduction listing the ways in which this work replicates vs. extends the previous paper by Weiss et al.

We thank the reviewer for prompting clarity on the respective purpose of each study. The previous paper by Weiss et al., 2021 focused on the physiological analysis of evidence integration within a sequence. We pooled all trials and focused our analyses on the comparison between how participants process evidence when it is a cue (uncontrollable condition) vs. when it is an outcome (controllable condition). We did not focus on the differences between repeat and switch trials in this previous paper.

In sharp contrast to this previous study, here, we specifically focus on response switches, when participants change their mind about the current hidden state. This focus allows us to study and characterise a moment in which participants seek information in the task. Importantly, for behavioral analyses, we no longer analyse what happens during the course of a sequence of stimuli, we instead take into account the cumulative evidence brought by the whole sequence. Our physiological analyses then focus on studying the contrast between repeat and switch decisions between experimental conditions so as to characterise the neural basis of information seeking. We now clarify the goal of each study in our introduction:

“We previously used these experimental conditions to compare how participants integrate evidence when it is a cue (uncontrollable condition) vs. when it is an outcome (controllable condition) (Weiss et al., 2021). Here, we focus on the comparison of repeat and switch decisions between these conditions so as to isolate the behavioral signatures and neural basis of information seeking, while replicating most of the previously observed effects in Weiss et al. on behavioral choices and their computational modeling (Weiss et al., 2021).” (p. 4)

6) Overall, the introduction is a little thin and would benefit from being expanded. First, the statement that "information-seeking has been mostly studied under 'exploration-exploitation' dilemma is not true – there are many recent studies that have studied information-seeking in humans using other paradigms. Second, and most importantly, the introduction currently lacks the rationale for the proposed work – in addition to dissociating information-seeking from changes-of-mind, which is more of a methodological aim, what are the key questions that the authors trying to address, and what are the hypotheses given the current literature? Finally, the use of MEG and other physiological measures (pupillometric, cardiac patterns) is not motivated at all in the introduction. The introduction should set the stage as to why collecting these data is needed given the question of interest. Similarly, the hypotheses should also set the stage for the specific analysis choices performed later, i.e. in what way does each presented analysis answer the question?

We thank the reviewer for prompting clarity on the purpose of the study, the motivation for our hypotheses, and the conceptual advances on information seeking processes that the study allows.

First, we agree with the reviewer that other important previous work has studied information seeking outside of exploitation-exploration dilemmas, which we now review in more depth. In particular, we have identified several families of paradigms tested in human and non-human primates which we now unpack in our introduction, highlighting the similarities and differences with the present paradigm:

1. A first family of information seeking paradigms are belief updating tasks in which human participants are presented with statements about life events or self-relevant information or judgements from others, and are asked whether they would like to see more information about this (e.g., Sharot et al., 2011 *Nat Neurosci*; Kelly and Sharot, 2021 *Nat Commun*; Gesiarz et al., 2019 *Plos comput biology*). Another “intermediate” study has manipulated not actual controllability, but instructed controllability, i.e., what participants are told about their degree of control (Stolz et al., 2020 *Nature communications*). Typically, these trials are not decision-making under uncertainty but are one-shots, and do not require learning or adaptation unlike in our study and in more continuous and open-ended ecological environments in which information seeking usually occurs. These decisions are described (in the sense of Hertwig and Erev, 2009) with probabilities or uncertainty depicted directly without ambiguity. Another important difference is that the nature of the information proposed varies in its valence and desirability (e.g., Hertwig et al., 2021 *Psychology and Aging*).

2. A second family of paradigms relies on perceptual decision-making as a model system to study the cognitive and neurobiological mechanisms underlying perceptual changes-of-mind (e.g., Desender et al., 2018 *Psychological Science*; Fleming et al., 2018 *Nat Neurosci*; Rollwage et al., 2020 *Nat Commun*). In these studies, expected uncertainty typically comes from the same source and is manipulated as the amount of sensory noise present in decision evidence. All trials and independent and there is no notion of volatility or reversals to adapt to – no unexpected uncertainty is manipulated.

3. Non-instrumental information-seeking has been the focus of recent studies in non-human primates (Monosov and Rushworth, 2022 *Neuropsychopharmacology*). In these studies, typically non-human primates are presented with visual cues predicting upcoming rewards, and acquiring information about these cues presents no immediate or direct instrumental value to the animal (e.g., White et al., 2019 *Nat Commun*). These paradigms also include no notion of volatility, with independent trials (no hidden state to track).

4. In decision-making under uncertainty, finally, a common family of paradigms is bandit tasks in which participants are asked to sample between two or more slot machines so as to maximize gains or rewards, and/or avoid losses or punishments, leading to exploite-explore dilemmas (e.g., Daw et al., 2006 *Nature*; Zorowitz and Niv, 2021 *BioRxiv*; Collins and Frank, 2012 *Eur J of Neurosci*), or other forms of economic decision-making (Kaanders et al., 2021 *J Neuro*; Wilson et al., 2014 *J Exp Psychol Gen*), and foraging (e.g., Kolling et al., 2012 *Science*). The present paradigm follows this line of research but critically moves away from it and explore-exploit dilemmas and instead focus on isolating information seeking from exploratory decisions, i.e., when people change their mind about the reliability of the current strategy and go explore something else.

These previous families of paradigms vary on many dimensions, in particular the sources of uncertainty sources and the type of stimuli manipulated, which has made their findings’ direct comparison difficult. Critically, none of these previous paradigms has directly manipulated controllability over decision evidence (information acquired). The unique contribution of the present study is to isolate information seeking by contrasting evidence sampling under controllable and uncontrollable conditions, all other decision features such as stimuli and levels and forms of uncertainty being equal.

Second, the main goal of the study in isolating information seeking from changes-of-mind is much more fundamental than a methodological goal. We have now reworked Figure 1 and included a new panel to explain the different concepts at play in the study. Current protocols proposed to study information seeking in the context of exploitation-exploration trade-offs have important confounds, in that information seeking is virtually always associated with changes-of-mind, meaning that different cognitive steps co-occur that are not specific to information seeking per se. It is an important conceptual distinction that has not been addressed by previous work (although for an exception, see Collins and Koechlin, 2012 *Plos biology*).

Third, we agree with the reviewer that our physiological hypotheses should be presented early on and we now better motivate our physiological analyses in our introduction section. We sought to evaluate whether there were additional correlates of subjective uncertainty associated with the process of information seeking. Because a switch response engages participants’ sense of agency in the C+ but not in the C- condition, we expected this form of evidence sampling to engage different networks, but it has not been examined in previous work whether these different forms of information seeking rely on separate or similar brain systems.

We have now rewritten our introduction to reflect this larger background (p. 3-4):

“In these paradigms, participants evolve in controllable environments and usually sample one among several options to maximize reward. Therefore, they have to either exploit a currently rewarding option, or sacrifice rewards to explore alternative options and seek information about their possible rewards (Rich and Gureckis, 2018; Wilson et al., 2014). This trade-off means that in these paradigms, exploration differs from exploitation not only in terms of information seeking, but also in terms of other co-occurring cognitive events, including overt response switches and covert changes-of-mind.

These different families of paradigms developed for studying information seeking vary on several dimensions, particularly the sources of uncertainty (Fleming et al., 2018), the stimuli used (Gesiarz et al., 2019), the desirability of the information to be sought (Hertwig et al., 2021), and the degree of control over information sampled (Desender et al., 2018). These differences have made direct comparisons between paradigms extremely challenging. To date, no study has directly manipulated control over evidence sampling in otherwise aligned experimental conditions. At the neurophysiological level, exploration is known to be associated with larger pupil-linked arousal (Jepma and Nieuwenhuis, 2011) and increased activity in lateral prefrontal regions in electroencephalographic (EEG) and blood oxygen-level dependent (BOLD) activity (Donoso et al., 2014; Tzovara et al., 2012). However, due to confounds between information seeking and other cognitive events during exploration in these studies, it remains unclear whether information seeking is associated with specific neurophysiological signatures.”

Methods7) Was the model fitted to all trials together, or separately for each condition?

We have now clarified this point in our Methods section (p. 29):

“All trials were fitted together, but each parameter was allowed to vary between conditions, and we compared the best-fitting parameters using paired t-tests at the group level.”

8) The model validation is rigorous, but would the same patterns also be observed out-of-sample? Since only one model is tested, there is always a risk of overfitting, which could be addressed with out-of-sample validation.

Here, our goal was not to arbitrate between models, but to compare our two conditions under the same model, with parameters between conditions under the same model. This is why we have fitted the same model in both our conditions. We have selected the most parameterised model, with all the free parameters allowed to vary between conditions. Even if we were to be overfitting, this is not an issue here, because our statistics are based on the comparison between best-fitting parameter values across C+ and C- conditions. Overfitting does not inflate the probability of observing significant differences in parameter values across conditions. This is important in order to avoid that parameters’ estimates may be contaminated by variance that is unrelated to the cognitive process each parameter aims to capture. Our approach here is similar to incorporating regressors of no interest in a regression, so as to avoid regressors of interest to capture unrelated variance.

In addition, the behavioral effects associated with each of our parameters at the group level indicate that overfitting is unlikely here. If we were overfitting, meaning that if one of the parameters was essentially capturing noise, we would not observe consistent effects of a given parameter across participants, as is illustrated in Figure 4 supplement 3 (choice parameters) and Figure 4 supplement 4 (confidence parameters). For example, when we do a median split of participants according to their best-fitting perceived hazard rate, we observe a translation in the repetition curve indicating that the lower the hazard rate, the more inconsistent evidence is needed for participants to change their mind. Had hazard rate captured noise in case of overfitting, we would not observe such a consistent effect at the group level.

In sum, our assumption that overfitting is very unlikely to have occurred is validated by the fact that each parameter is specifically associated with a given consistent behavioral pattern (Figure 4 supplement 3 and 4), which would not happen if any of these parameters was capturing noise as in the case of overfitting. We now provide more information about our model validation approach and the risk of overfitting in our Methods section (p. 30):

“Here, we selected the most complete (parameterized) model, even at the risk of overfitting, because our goal was not to arbitrate between different models, but to compare our two conditions and our parameters under the same model. In addition, the behavioral effects associated with each of our parameters at the group level indicate that overfitting is unlikely to have occurred (see ‘Model validation’ below). Indeed, if overfitting had occurred, meaning that if one of the parameters was essentially capturing noise, we would not observe consistent effects of a given parameter across participants (Figure 4 supplement 3 and 4).”

9) Please provides the statistics for the validation data presented on p12 (Figure S2 and S3)

Figure S2 (now Figure 2 supplement 3) provides model simulations from individual best-fitting parameters. Upon the reviewer’s suggestion, we have now performed a full psychometric analysis of these simulation data. Note that the psychometric analysis of model behavior is not as central to our conclusions as that of participants’ behavior. Indeed, the purpose of model validation is to ensure that the model reproduces key *signatures* or behavioral *patterns* present in the human data (Palminteri et al., 2017 *TICS*; Wilson and Collins, 2019 *eLife*), which our model does very well. For the sake of concision, we have therefore included the statistical values for this full psychometric analysis directly into the figure, but we report them below for completeness.

We observed that the model was slower to adapt after a reversal in the C+ condition, with psychometric fits indicating a higher reversal time constant in this controllable condition (*t*_32_=-6.0, *p*=9.5×10^-7^). The model behavior also reached a higher asymptotic reversal rate in the C+ condition (*t*_32_=-6.6, *p*=2.0×10^-7^). The model’s simulated confidence decreased after a reversal in both conditions. However, confidence decreased more sharply in the C+ condition as indicated by psychometric fits of a confidence ‘drop’ parameter (*t*_32_=5.22, *p*=1.0×10^-5^). The confidence time constant characterizing the slope of confidence increase after a reversal was similar across conditions (paired *t*-test, *t*_32_=-0.50, *p*=.62).

We found a stronger choice-PSE in the C+ condition, as in human data (*t*_32_=-8.3, *p*=1.7×10^-9^), together with a similar sensitivity to the evidence, as in human data (*t*_32_=0.79, *p*=0.44). Finally, we found a significant confidence-PSE difference between conditions (Wilcoxon signed rank test, *z=*-3.59, *p*=.00033), indicating that the model needed more inconsistent evidence to be equally confident in switch and repeat decisions in the C+ condition, like humans. We also found that the sensitivity of confidence reports to evidence (slope) was slightly smaller in the C+ than in the C- condition on switch decisions, albeit only borderline significant (*t*_32_=1.86, *p*=0.07).

Figure S3 (now Figure 4 supplement 3 and 4) displays the effects of parameters on the four main behavioral signatures, namely, reversal and repetition curves for choice and confidence, from which we only draw qualitative conclusions. The purpose of these figures is only to visualise the selective effects of each of our free parameters. We do not perform any group inference or statistical analyses at the group level from these analyses. We now make this point clear in our Methods section (p. 30-31):

“We also performed a median split across participants on the best-fitting parameter values (Figure 4 supplement 3 and 4). We then averaged simulations of the model for each subgroup, to further illustrate the independent contribution of each of these parameters. Note that we do not draw group inferences from these median split analyses – the goal is to visualise the effect of each of the parameters qualitatively.”

10) How recoverable are the parameters if the model is re-fit to simulated data?

We have now performed a full parameter recovery analysis, as described in our Methods section (p. 29):

“To ensure that our fitting procedure was unbiased, we performed a parameter recovery analysis (Figure 4 supplement 1). We simulated choice and confidence sequences using generative parameters randomly sampled in a uniform distribution between the minimum and maximum of participants’ best-fitting parameter values in each condition. This procedure ensures that generative parameters were independently sampled. We then fitted these data using the same fitting procedure as for participants (except with three instead of five random starting points) and calculated the correlations between generative and recovered parameters, presented in a confusion matrix (Figure 4 supplement 1).”

We now provide these results in a new Figure 4 supplement 1.

We found a satisfactory parameter recovery, indicating that our fitting procedure was reliable, as explained in our Results section (p. 12):

“Importantly, we also established that our fitting procedure provided a satisfactory parameter recovery (see Methods). All correlations between generative and recovered parameters were high (all rho>0.78, all p<1014), while other correlations were low as indicated in a confusion matrix (Figure 4 supplement 1).”

11) Are parameters correlated with each other? The authors should provide a correlation table of the parameters in the supplement.

We have now inspected the correlation between our five model parameters. Apart from a significant negative correlation between inference noise and hazard rate (rho=-0.51, *p*=.0025), and a borderline negative correlation between confidence threshold and hazard rate (rho=-0.36, *p*=.0409), no other correlations were significant (all rho<0.31, all *p*>.074) (*N*=33 participants, individual best-fitting parameters averaged between conditions). We now provide a correlation matrix of parameters as a new Figure 4 supplement 2:

We also provide these findings in our Results section and comment on their implications (p. 14):

“We further validated the independent role of each parameter in two ways. First, we examined correlations between best-fitting parameters across participants (Figure 4 supplement 2). We found a significant negative correlation between inference noise and hazard rate (rho=-0.51, p=.0025), in line with a previously reported trade-off between these two sources of noise (Weiss et al., 2021). We also found a borderline correlation between confidence threshold and hazard rate (rho=-0.36, p=.0409). However, all other correlations were not significant (all rho<0.31, all p>.074), indicating that each parameter captured independent portions of the variance (Figure 4 supplement 2). Second, we did a median-split of participants into groups of high and low parameter values (Figure 4 supplement 3 and 4), each parameter having a selective influence on choices and confidence. Even when parameters were similar across conditions (e.g. confidence gain), there was still a substantial inter-individual variability that had a visible effect on participants’ confidence, indicating the necessity of each parameter in capturing qualitative signatures participants’ choices and confidence. Although the model allows for potentially distinct effects between inference noise and hazard rate, here, our participants happened to be distributed along a particular regime, with no independence between these two parameters, as visible in similar effects of lower inference noise and higher hazard rate on behavior (Figure 4 supplement 3), in line with their negative correlation.”

“We examined correlations of the best-fitting parameters averaged across conditions between participants (Figure 4 supplement 2)”. (Methods section, p. 29)

Results12) It would help to preview the specific diagnostic effects that isolate the process of interest, and then to repeat those in the model fit. Currently, we see four effects in Figure 2, four effects in Figure 3, and three effects in Figure 4. When we get to Figure 5D, slightly different effects are presented in comparison to the model. This was quite confusing and more clarity is required about the specific behavioral pattern of interest. This may just be a matter of slight reordering: the supplement has clear model-data matches, and placing those in the main figure (to highlight that most of the basic effects in Figures2-4 are indeed explained by the model) would allow a better appreciation of where the model fails to capture human behavior.

We thank the reviewer for prompting clarity on the specific behavioral signatures that are key to diagnose the model validity and have now fully reorganised our figures accordingly. We have now grouped ex. Figure 2 and Figure 3 into a single figure with the four key curves and the corresponding psychometric parameters (new Figure 2). To allow for an easy comparison between human and model behaviours, we now have the model validation in a figure aligned with the same signature patterns.

Upon the reviewer’s suggestion, we have separated the ex. Figure 5C and 5D in a new separate Figure 5 to isolate where the model fails to capture human behavior. We hope that this reorganisation now allows to appreciate the similarities between model and human behavior and between experiments.

13) (see also (2) above) The data in Figure S1 are a crucial control, but it is not yet clear that there is no substantial difference in pro- vs. retro-spective decisions, which might (partially) explain the Cb vs. Ob differences. If controllability was the only important factor for behavioral effects, wouldn't we expect no difference between conditions in Figure S1? This would require a change in the authors' strength of interpretation as specifically pertaining to control.

We agree with the reviewer that the strength of our conclusions on the nature of the difference between the C- and C+ conditions, whether controllability or temporal orientation (prospective vs. retrospective), need to be aligned with the evidence provided and we now nuance our interpretations. We refer the reviewer to our detailed response to the main comment 2.

Briefly, we now also present the choice data in addition to the confidence data from Experiment 3. Critically, the findings of Experiment 3 reveal that the nature and magnitude of the contrast between retrospective and prospective conditions is different from the contrast between the original C- and C+ conditions, and therefore cannot account for them as an overall explanation of the results; however, a remaining effect of temporal direction of the inference might still be present in the C- vs. C+ conditions, which we now acknowledge:

“Furthermore, we sought to validate controllability as the true cause of differences between conditions. In Experiment 3, we examined whether a distinction in temporal focus (prospective instead of retrospective inference in an uncontrollable context) would account for the differences between the original conditions (Figure 2 supplement 1). Although the pattern of choices was markedly different, for confidence it remains possible that a lingering effect of temporality affected the original conditions, even if it cannot account for the results overall.” (Discussion section, p. 23)

14) Do the analyses in Figure 4 control for the difference in objective evidence (logL per trial)?

We controlled indeed for the strength of evidence provided by the sequence of stimuli in Figure 2G-2H for the data presented in (ex)Figure 4A (now Figure 3). In other words, Figure 4A presents the same data as Figure 2G but pooled across evidence strength levels. However, because in Figure 4A predictors would be binary, i.e., repeat and switch, and C- and C+ conditions, we considered that an ANOVA was the most relevant and appropriate test here. Unfortunately, too few events per cell are included in Figure 4B and 4C to further be able to do a breakdown as a function of evidence strength in bins, as we did for confidence in Figure 4A. Moreover, it is worth noting that the observed effects are in the opposite direction of what we would expect if evidence strength was a confound. Indeed, we found that in the C+ condition, participants need more evidence to switch, which should lead to higher confidence (more evidence => more confidence). Instead, we found a *reduced* confidence on switch decisions in the C+ condition.

Regarding Figure 4B and 4C (now Figure 3B and 3C), these analyses do not control for variations in objective evidence. Data for these analyses cannot be further split further as a function of objective evidence due to a low number of events corresponding to each case (cell) studied: as expected, there was not a homogenous amount of evidence leading all of these events. For instance, participants would rarely confirm a change-ofmind with high confidence if it was based on very little objective evidence.

Note, however, that by design, the objective evidence level was matched across conditions on average, and that the PSE psychometric analyses take into account the objective evidence – instead, Figure 4 focuses on the consequences of response switches on participants’ subsequent behaviour.

We now clarify this point in our Methods section (p. 29):

“In a 2 × 2 repeated measures ANOVA, we examined the influence of RESPONSE TYPE (repeat, switch) and CONDITION (C-, C+) on the fraction of high confidence responses (Figure 3A, which corresponds to the same data as Figure 2G pooled over objective evidence levels). For switch trials, we further examined confidence on switch trials that were confirmed on the next trial (“change-of-mind confirmed”) as compared to switch trials after which participants went back to their previous response (“change-of-mind aborted”; Figure 3B). Finally, we examined the fraction of changes-of-mind confirmed (over all changes-of-mind) as a function of whether the change-of-mind was done with high or low confidence (Figure 3C). These analyses are pooled over objective evidence levels due to each of these events not being distributed homogeneously across evidence levels.”

Upon the reviewer’s suggestion, we have now performed a logistic regression to take into account strength of evidence. Specifically, we included the trial-by-trial evidence strength in favour of the previous response as a co-regressor:

Confidence(t) (high/low) ~ Response type(t) (repeat/switch) + Condition(t) (C-/C+) + Response type(t)*Condition(t) + Evidence level(t)

and

Confidence(t) (high/low) ~ change-of-mind(t) (confirmed/aborted) + Condition(t) (C-/C+) + change-ofmind(t)*Condition(t) + Evidence level(t)

We implemented a regularised logistic regression with gaussian priors on each regression coefficient (mean=0, SD=2) in order to account for the fact that some of our participants have few events per cell, which otherwise led to unreliable regression coefficient estimates.

The results are in line with our initial model-free analysis (original Figure 4A). Namely, repeat trials led to higher confidence (*t_32_*=7.66, *p*=9.8×10^-9^), confidence was higher in the C- than in the C+ condition (*t_32_*=4.05, *p*=.00029), with a significant interaction between Response type and Condition factors (*t_32_*=-4.43, *p*=.0001), while controlling for evidence level in the same regression model, which positively contributed to confidence, as expected (*t_32_*=14.69, *p*=8.8×10^-16^) (Figure 3 supplement 1A).

Likewise, in a logistic regression we have controlled for evidence strength in the analysis relating confidence to whether the change-of-mind of the previous trial was confirmed or aborted (related to original Figure 4B). The results confirm model-free observations, with participants overall being more confident when they confirm than when they abort a change-of-mind (*t_32_*=5.49, *p*=4.6×10^-6^), also more confident in the C- than in the C+ condition (*t_32_*=3.92, *p*=.0004), with a significant interaction between these factor (*t_32_*=-3.38, *p*=.0019), while controlling for evidence level (*t_32_*=11.9, *p*=2.49×10^-13^) (Figure 3 supplement 1B).

We also computed, for each of the cases, the average evidence quantity in favour of the participant’s previous response, and compared the obtained interactions patterns to those obtained in the original Figure 4 (Figure 3 supplement 1C-D). If the interaction patterns are different from the original results, it means that the original results cannot be explained away by differences in evidence level.

In a 2 × 2 ANOVA on mean evidence level, as expected, we found a main effect of RESPONSE TYPE, F_1,32_=147.8, *p*=1.6x10^-13^, because negative evidence will lead participants to switch from their previous response. We found a higher overall evidence level in the C+ than in the C- conditions (main effect of CONDITION, F_1,32_=6.19, *p*=.018). We observed a significant interaction between RESPONSE TYPE and CONDITION (F_1,32_=57.1, *p*=1.3x10^-8^), indicating that in the C+ condition, participants have seen more evidence against their previous choice when they switch their response than in the C- condition, in line with choice-PSE results. Having seen more evidence, we would expect them to be *more* confident, whereas they actually are *less* confident in that condition (original Figure 4A). Therefore, these findings provide further evidence that the strength of evidence does not explain away our results on confidence in response switches.

Moreover, evidence levels were higher on trials in which switch decisions were confirmed as compared to aborted (F_1,32_=264.2, *p*<.001). There was no main effect of CONDITION (F_1,32_=0.67, *p*=.41), with an interaction between CONDITION and CONFIRM-ABORT factors (F_1,32_=42.6, *p*=2.3x10^-7^) (Figure 3 supplement 1B). This pattern reflects that evidence level partly contributes to confidence reports, but is markedly different from the patterns observed for the fraction of high confidence responses (original Figure 4B).

In the case of original Figure 4C, we have displayed proportions, namely, the fraction of changes-of-mind confirmed out of the total changes-of-mind performed with high confidence and low confidence respectively. For this reason, we cannot compute a mean evidence level in each of the four cases in original Figure 4C. Instead, to take into account evidence level, we can compare the mean evidence level on (i) changes-ofmind performed with high confidence and then confirmed, (ii) performed with low confidence and then confirmed, (iii) performed with high confidence and then aborted, and (iv) performed with low confidence and then aborted, separately for the two conditions so a total of 8 cells. However, data for these analyses cannot be further split further as a function of evidence due to a low number of events corresponding to each case (cell) studied: as expected, there was not a homogenous amount of evidence leading all of these events. For instance, participants would rarely confirm a change-of-mind with high confidence if it was based on very little objective evidence. Likewise, a logistic regression would not be appropriate if we have too few data points, as in the case of the original Figure 4C. Note, however, that by design, the objective evidence level was matched across conditions on average, and that the PSE psychometric analyses take into account the objective evidence – instead, Figure 4 focuses on the consequences of response switches on participants’ subsequent behaviour.

We now clarify this point in our Methods section (p. 29):

“In a 2 × 2 repeated measures ANOVA, we examined the influence of RESPONSE TYPE (repeat, switch) and CONDITION (C-, C+) on the fraction of high confidence responses (Figure 3A, which corresponds to the same data as Figure 2G pooled over objective evidence levels). For switch trials, we further examined confidence on switch trials that were confirmed on the next trial (“change-of-mind confirmed”) as compared to switch trials after which participants went back to their previous response (“change-of-mind aborted”; Figure 3B). Finally, we examined the fraction of changes-of-mind confirmed (over all changes-of-mind) as a function of whether the change-of-mind was done with high or low confidence (Figure 3C). These analyses are pooled over objective evidence levels due to each of these events not being distributed homogeneously across evidence levels.”

We also include these new logistic regressions controlling for evidence level in our revised manuscript:

“To take into account any effects of evidence strength in these change-of-mind analyses, we further performed two logistic regressions (Figure 3 supplement 1A-B). First, we aimed to predict confidence using Response type, Condition, their interaction, and trial-by-trial evidence strength in favour of the previous response as a co-regressor (related to Figure 3A). Second, we aimed to predict confidence using change-ofmind (confirmed vs. aborted), Condition, their interaction, and trial-by-trial evidence strength in favour of the previous response (related to Figure 3B). We implemented regularised logistic regressions with gaussian priors on each regression coefficient (mean=0, SD=2) in order to account for the fact that some of our participants have few events per cell, which otherwise led to unreliable regression coefficient estimates. Group-level significance of regression coefficients was assessed using one-sample t-tests. We also computed the average evidence quantity in favour of the participant’s previous response, and compared the obtained interactions patterns to those obtained in the change-of-mind analyses in Figure 3 (Figure 3 supplement 1C-D).” (Methods section, p. 31)

“A logistic regression confirmed these results (see Methods). Repeat trials led to higher confidence than switch decisions (t_32_=7.66, p=9.8×10^-9^), confidence was higher in the C- than in the C+ condition (t_32_=4.05, p=.00029), with a significant interaction between RESPONSE TYPE and CONDITION (t_32_=-4.43, p=.0001), while controlling for evidence level in the same regression model, which positively contributed to confidence, as expected (t_32_=14.69, p=8.8×10^-16^) (Figure 3 supplement 1A).” (Results section, p. 11)

“A logistic regression confirmed these results (see Methods), with participants overall being more confident when they confirm than when they abort a change-of-mind (t_32_=5.49, p=4.6×10^-6^), also more confident in the C- than in the C+ condition (t_32_=3.92, p=.0004), with a significant interaction between these factor (t_32_=-3.38, p=.0019), while controlling for evidence level (t_32_=11.9, p=2.49×10^-13^) (Figure 3 supplement 1B). Moreover, while these results indicate that evidence partly contributes to confidence, as expected, the patterns of evidence strength were markedly different from those of changes-of-mind (compare Figure 3A-B to Figure 3 supplement 1C-D).” (Results section, p. 12)

15) It is frequently stated that the Ob condition bestows instrumental control to participants over the sampling of stimuli. These statements should be modified, given that participants are told which category of stimulus they must generate. They do not have complete control over which stimuli they sample, unless they are willing to upset the experimenter!

The reviewer is correct that the control is indeed over the category of stimuli produced, not over each stimulus in the sequence itself. In other words, participants have control / can decide over which stimulus to sample (left or right action in the controllable condition), although they do not have control over the target category, i.e., the generative category from which samples are drawn.

Controllability is a property of this C+ (ex. Ob) condition, in which participants have a goal, just like in classic n-arm bandit task. It is not a forced choice but they are asked to produce stimuli from a particular target category; in that sense, it is not a pure free sampling but goal-directed sampling. We have now clarified this point in our Methods section (p. 27):

“In the controllable (C+) condition, participants were instructed to draw stimuli from a given category (Figure 1C). An instruction screen indicated the target color category for each block (counterbalanced across blocks). In the C+ condition, participants have control over which stimuli to sample, but are not fully freely sampling, since they asked to produce stimuli from a target color category.”

16) It is not clear what action participants perform to start the trial in the Ob condition. This needs including. In outlining this it would also be great to include consideration of potential overlaps between the responses at the start and end of the trial in Ob, and whether this might have played into the effects.

In both conditions, participants perform exactly only one motor action per trial. In the very beginning of a block of trials of the C+ condition, participants are asked to perform a random action to initiate the very first sequence of stimuli, as they have no idea yet of what the current mapping of categories is. In the C- condition, similarly, participants are asked to initiate the block of trials by performing a random action. Therefore, the first response required before having seen any sequence is random in both conditions, so that motor actions are precisely aligned between the C+ and C- conditions.

We now clarify this point in our Results (p. 5) and Methods (p. 26-27) sections:

“In-between each sequence, participants were asked to provide a response regarding the current hidden state of the task, together with a binary confidence estimate in their response (high or low confidence) using four response keys (Figure 1B). […] In both conditions, participants perform one action per trial.” (p. 5)

“The conditions were otherwise fully symmetric, tightly matched in terms of visual and motor requirements. In both conditions, participants perform one action per sequence (after seeing a sequence in the C- condition, before seeing a sequence in the C+ condition).” (p. 26-27)

17) Please add the change-of-mind data (i.e. equivalent of Figure 2A-B and 3A-B) for experiment 3, so it is possible to dissociate which effects are unique to the main experiment and which effects may be due to the temporal direction of the decisions. It is important to clarify this distinction as it can actually help narrow down the unique effect of controllability over temporality.

We have now added the choice psychometric analysis (equivalent of ex Figure 2A-B, adapted from Weiss et al., 2021) and the confidence psychometric analysis (equivalent of ex Figure 3A-B) in a new Figure 2 supplement 1 (printed in response to comment 2). We fully agree with the reviewer about the importance of isolating the unique effect of controllability over temporality using Experiment 3, and we refer the reviewer to our responses to comment (2) that addresses this point in depth.

18) It would be useful to provide follow-up analyses to determine how much of the behavioral results can be explained by the difference in prior belief reported on p19, as well as discuss how much of these findings are then more consistent with a role of controllability per se, rather than a role for motivated beliefs/confirmation bias due to the presence of a target.

We agree with the reviewer that these notions are related, and we now provide a number of pieces of evidence in our data that speak against an interpretation in terms of confirmation bias or motivated beliefs.

First, by using a computational model, we seek to provide a mechanistic explanation for our observed behavioral effects. Specifically, stronger prior beliefs are actually what will *produce* the effects we see. Controllability is only a high-level notion manipulated by the experimenter, while the computational model has no notion of controllability built into it. But by fitting the computational model to participants’ data in the two controllability conditions, the model happens to explain participants’ choices and confidence by the presence of stronger prior beliefs in the controllable (C+) condition. It is those stronger prior beliefs that produce behavior, not the other way around.

Second, Experiment 2B also addresses this point to some extent. In Experiment 2B, the target goal is not constant during a block and varies between trials. This means that if confirmation bias were an explanation for the findings, the direction of the confirmation bias would change from one trial to the next, and its impact on behaviour should therefore decrease.

Third, at the neural level, in our previous study, we have argued that “the strength of belief-inconsistent evidence could be decoded [from MEG signals] with equal precision across conditions”, something at odds with a confirmation bias which would decrease the weighting of belief-inconsistent evidence. Instead, we found an absence of difference between the neural coding of belief-consistent and belief-inconsistent evidence.

Fourth, at the behavioral level, the reviewer is correct that a confirmation bias in the C+ would be associated with an increased choice-PSE, which we observe in the data. Confirmation bias has previously been defined as a selective weighting of belief-consistent evidence (Bronfman et al., 2015 *Proc R Soc B*; Peters et al., 2017 *Nat human behaviour*; Talluri et al., 2018 *Current Biol*).

However, a confirmation bias in the C+ condition would also be accompanied by a significant decrease in the sensitivity to the objective evidence in the C+ condition, an aspect that is absent from the human data (Figure 2). Indeed, a confirmation bias would predict that in the C+ condition, the evidence used by the participant is not the objective evidence, but a biased subjective representation of the evidence where the inconsistent evidence is underweighted relative to the consistent evidence. Hence, sensitivity to the objective evidence (measured as our psychometric parameter) would be decreased in the presence of a confirmation bias in the C+ condition. In other words, we have considered the objective evidence with respect to participants’ previous response (Figure 2E-H) as a key relevant variable for decision-making, but if participants use irrelevant or wrong variables (e.g., confirmation bias), it should have produced a degraded sensitivity to the objective evidence (Beck et al. 2012 *Neuron*).

To falsify more directly a confirmation bias explanation, we have conducted a new analysis. We fitted the differences in response reversal and response repetitions curves (Figure 2) between the C- and C+ conditions in Experiment 4 using two alternative accounts: 1. a change in perceived hazard rate in the C+ condition, and 2. a confirmation bias in the C+ condition (controlled by a gain parameter assigned to belief-inconsistent evidence). As expected, a change in perceived hazard rate produced a minimal change in the sensitivity to evidence which matched human data (interaction with participants: F_1,23_=0.4, *p*=0.54). In contrast, a confirmation bias produced a substantial decrease in the sensitivity to evidence in the C+ condition which did not match human data (interaction with participants: F_1,23_=10.0, *p*=.004). This interaction ‘falsifies’ (in the meaning of Palminteri et al., 2017 *TICS*) the most standard account of confirmation bias.

We now provide evidence to rule out this alternative explanation in terms of “confirmation bias” in our Discussion section (p. 23-24):

“Despite stimuli carrying no explicitly affective or rewarding value, it remains possible that the mere presence of a target in the C+ condition makes the target category desirable, and may produce a form of confirmation bias (Talluri et al., 2018). However, at the behavioral level, a confirmation bias would predict that in the controllable condition, the sensitivity to evidence should be degraded when the sequence of evidence is inconsistent with participants’ previous choice, a prediction that is absent from human data (Figure 2). At the neural level, a confirmation bias would decrease the weighting of belief-inconsistent evidence. Instead, in our previous study, we found an absence of difference between the neural coding of belief-consistent and belief-inconsistent evidence (Weiss et al., 2021). In addition, the findings of Experiment 2B, where the target category changes from trial to trial, also mean that the differences between conditions are unlikely to reflect a bias in direction of the target category (Gesiarz et al., 2019). Indeed, the direction of this bias would change from one trial to the next, and should therefore decrease in Experiment 2B – which we did not observe.”

Discussion19) There is some concern about the interpretation of the results – in particular, whether the observed differences between the cue-based and outcome-based conditions could be better explained by the presence of a target-induced confirmation bias in the outcome-based condition, which would induce a strong motivation to confirm that the chosen action does lead to the target. In other words, it is possible that having a target (e.g. "draw orange") may bias subjects towards believing they are drawing from that target even when they are not, and as a result needing less evidence to reach that conclusion (similar to the effect shown in Gesiarz, Cahill and Sharot, 2019, Plos Computational Biology). This could in turn lead to the observed patterns of results, i.e. needing more evidence against orange to switch, being less confident when switching etc. Additionally, the result that prior beliefs are stronger in the Ob condition (p18-19) is consistent with this idea, since confirmation bias is usually associated with stronger prior beliefs.

We would like to refer the reviewer to the series of arguments in response to comment (18), which we hope have thoroughly addressed the alternative explanation of a confirmation bias producing our pattern of results.

We also note that the types of motivated beliefs in the paradigm of Gesiarz et al., 2019 are beliefs that are associated with desirable states: one prefers to be in a state with greater rewards than losses rather than the other way around. In contrast, here, participants had no particular reason to be more motivated that the blue or the orange category of perceptual stimuli is the current hidden state. Moreover, no reward or losses were manipulated explicitly, for instance using monetary outcomes. No notion of valence or affective value is different between the C- and C+ conditions, or between the blue and orange generative categories, unlike in other confirmation bias paradigms where stimulus statements carry a strong affective content (e.g., Sharot et al., 2011 *Nat Neurosci*).

Nevertheless, it remains possible that the mere presence of a target category in the C+ condition could constitute a goal and create desirability to find more evidence in favour of that target category (Castagnetti et al., 2021 *Sci Advances*); however, as detailed in response to the previous comment, this would have translated into specific effects which we do not observe.

We now unpack this point in our Discussion section (p. 24):

“Despite stimuli carrying no explicitly affective or rewarding value, it remains possible that the mere presence of a target in the C+ condition makes the target category desirable, and may produce a form of confirmation bias (Talluri et al., 2018). However, at the behavioral level, a confirmation bias would predict that in the controllable condition, the sensitivity to evidence should be degraded when the sequence of evidence is inconsistent with participants’ previous choice, a prediction that is absent from human data (Figure 2). At the neural level, a confirmation bias would decrease the weighting of belief-inconsistent evidence. Instead, in our previous study, we found an absence of difference between the neural coding of belief-consistent and belief-inconsistent evidence (Weiss et al., 2021). In addition, the findings of Experiment 2B, where the target category changes from trial to trial, also mean that the differences between conditions are unlikely to reflect a bias in direction of the target category (Gesiarz et al., 2019). Indeed, the direction of this bias would change from one trial to the next, and should therefore decrease in Experiment 2B – which we did not observe.”

20) The authors claim that there is a causal role for confidence in controlling changes-of-mind (p11). While this is an interesting idea, it is not clear that it can be fully evidenced in this task without an experimentally-controlled manipulation of confidence. The reason is that there could be a common cause to both high confidence and high propensity to confirm switch decisions without the two processes actually being causally related. One such common cause could be the strength of evidence. Therefore, this conclusion needs to be tempered and this issue mentioned in the discussion.

We agree with the reviewer about the existence of a common cause being possible. Note that we do take into account the strength of the evidence, one of such possible common causes, in our analyses on confidence (Figure 3, now Figure 2G-H). We would also like to refer the reviewer to our response to comment (14) and (xiii), in which we provide new analyses taking into account the strength of evidence, which shows consistent results. Here, in writing “a role for confidence”, we meant causality in a weak sense: the sense of temporal precedence. Yet, because we have not performed a direct causal manipulation of confidence here, we now have now tempered the claims associated with these findings in our Results section, paragraph ‘Separable effects of confidence and controllability on changes-of-mind’:

‘A possible role for confidence in changes-of-mind’ (title of Figure 3, p. 11)

‘We found a main effect of CONFIDENCE on the fraction of switches confirmed (F_1,27_=30.4, p=7.8×10^-6^), meaning that participants confirmed their switch more often when it was made with high confidence. This suggests a causal role for confidence in controlling changes-of-mind, even though we acknowledge that there was no experimentally causal manipulation of confidence here.’ (p. 12)

21) There is no discussion of alternative ways in which these data might have turned out. The discussion entirely reports claims and findings with which the present data are consistent. Are there no data or theories that could have led to alternative predictions? Currently the manuscript portrays the impression of such high consensus that there was little point running the studies, suggesting that the empirical patterns to date all point in identical directions. One possibility is that the stimulus ISIs are not very variable. Given participants can perceive better at certain oscillatory phases of visual processing (peaks), could they choose to start trials in Ob to align with peaks and thereby improve perceptual acuity? If this occurred, the influence must be weaker than those pulling in the opposite direction, but it could have led to an alternative outcome. If it's a straightforward analysis to perform/report, it may be interesting to see how the oscillations align with events differentially in Ob and Cb.

The purpose of the present study was to examine changes-of-mind in controllable and uncontrollable while otherwise comparable conditions, an endeavor that has not been examined before, to the best of our knowledge. It was not an easy or a given or trivial observation that the results turned out the way they have: indeed, information seeking has been mostly studied in separate fields of the decision-making literature, among which for instance:

(i) in the context of perceptual decision-making tasks, where participants are presented with evidence over which they have no control (e.g., Desender et al., 2018 *Psychological Science*);

(ii) in the context of belief update paradigms, where the information to be acquired is controllable and the proposed information typically varies in its desirability (e.g., Gesiarz et al., 2019 *Plos computational biology*); (iii) in the context of value-based decision-making tasks, where participants have control over options sampled (e.g., Daw et al., 2006 *Nature*).

It had not been investigated whether the different forms of information sampling at play would be similar or turn out to be different in controllable and uncontrollable environments, which we could investigate in the present paradigm. We have now highlighted this in our Introduction section (p. 3-4):

“In these paradigms, participants evolve in controllable environments and usually sample one among several options to maximize reward. Therefore, they have to either exploit a currently rewarding option, or sacrifice rewards to explore alternative options and seek information about their possible rewards (Rich and Gureckis, 2018; Wilson et al., 2014). This trade-off means that in these paradigms, exploration differs from exploitation not only in terms of information seeking, but also in terms of other co-occurring cognitive events, including overt response switches and covert changes-of-mind.

These different families of paradigms developed for studying information seeking vary on several dimensions, particularly the sources of uncertainty (Fleming et al., 2018), the stimuli used (Gesiarz et al., 2019), the desirability of the information to be sought (Hertwig et al., 2021), and the degree of control over information sampled (Desender et al., 2018). These differences have made direct comparisons between paradigms extremely challenging. To date, no study has directly manipulated control over evidence sampling in otherwise aligned experimental conditions. At the neurophysiological level, exploration is known to be associated with larger pupil-linked arousal (Jepma and Nieuwenhuis, 2011) and increased activity in lateral prefrontal regions in electroencephalographic (EEG) and blood oxygen-level dependent (BOLD) activity (Donoso et al., 2014; Tzovara et al., 2012). However, due to confounds between information seeking and other cognitive events during exploration in these studies, it remains unclear whether information seeking is associated with specific neurophysiological signatures.”

Regarding the alternative interpretation proposed by the reviewer, it is important to note that in both conditions, participants’ key presses trigger the presentation of the next sequence of stimuli after the same jitter delay in the two conditions. Therefore, participants could align their perceptual acuity with the stimuli equally well in the two conditions. Nevertheless, it is also important to note that stimuli are presented at 2 Hz, a rate that is sufficiently low to perceive stimuli properly. Here, the difficulty is to integrate, i.e., infer, the category source properly, despite perceiving stimuli easily. In other words, stimuli are of high contrast, they are not masked, and they are presented for sufficient time. Therefore, the cognitive bottleneck in our experiment is not on sensory processing, but on inference (see Drugowitsch et al., 2016 *Neuron* for an extensive decomposition of decision variability in terms of sensory, inference and action selection sources of variability).

We now clarify this point in our Methods section (p. 29):

“Stimuli were displayed at an average rate of 2 Hz with an inter-stimulus interval of 500 ± 50 ms. This rate is sufficiently low to perceive stimuli properly; here, the cognitive bottleneck was not on sensory processing, but on inference (Drugowitsch et al., 2016 Neuron). The last stimulus of each sequence was followed by a longer delay of 1000 ± 50 ms, before participants were probed for their response.”